# Synthesis and Characterization of Novel 2-Acyl-3-trifluoromethylquinoxaline 1,4-Dioxides as Potential Antimicrobial Agents

**DOI:** 10.3390/ph15020155

**Published:** 2022-01-27

**Authors:** Galina I. Buravchenko, Dmitry A. Maslov, Md Shah Alam, Natalia E. Grammatikova, Svetlana G. Frolova, Aleksey A. Vatlin, Xirong Tian, Ivan V. Ivanov, Olga B. Bekker, Maxim A. Kryakvin, Olga A. Dontsova, Valery N. Danilenko, Tianyu Zhang, Andrey E. Shchekotikhin

**Affiliations:** 1Gause Institute of New Antibiotics, 119021 Moscow, Russia; buravchenkogi@gmail.com (G.I.B.); ngrammatikova@yandex.ru (N.E.G.); ivanvi@yandex.ru (I.V.I.); 2Laboratory of Bacterial Genetics, Vavilov Institute of General Genetics, Russian Academy of Sciences, 119333 Moscow, Russia; maslov_da@vigg.ru (D.A.M.); sveta.frolova.1997@bk.ru (S.G.F.); vatlin_alexey123@mail.ru (A.A.V.); obbekker@mail.ru (O.B.B.); valerid@vigg.ru (V.N.D.); 3State Key Laboratory of Respiratory Disease, Guangzhou Institutes of Biomedicine and Health, Chinese Academy of Sciences, Guangzhou 510530, China; alam@gibh.ac.cn (M.S.A.); tian_xirong@gibh.ac.cn (X.T.); zhang_tianyu@gibh.ac.cn (T.Z.); 4China-New Zealand Joint Laboratory on Biomedicine and Health, Guangzhou 510530, China; 5Guangdong-Hong Kong-Macao Joint Laboratory of Respiratory Infectious Diseases, Guangzhou Institutes of Biomedicine and Health, Chinese Academy of Sciences, Guangzhou 510530, China; 6University of Chinese Academy of Sciences, Beijing 100049, China; 7Phystech School of Biological and Medical Physics, Moscow Institute of Physics and Technology (State University), 141701 Dolgoprudny, Russia; 8Institute of Ecology, Peoples’ Friendship University of Russia (RUDN University), 117198 Moscow, Russia; 9Organic Chemistry Department, Faculty of Natural Sciences, Mendeleyev University of Chemical Technology, 9 Miusskaya Square, 125190 Moscow, Russia; 10Chemistry Department, Faculty of Bioengineering and Bioinformatics, Lomonosov Moscow State University, 119991 Moscow, Russia; maxim.kryakvin@gmail.com (M.A.K.); olga.a.dontsova@gmail.com (O.A.D.); 11Center of Life Sciences, Skolkovo Institute of Science and Technology, 143028 Skolkovo, Russia; 12Shemyakin-Ovchinnikov Institute of Bioorganic Chemistry, 117997 Moscow, Russia

**Keywords:** 3-trifluoromethylquinoxaline 1,4-dioxides, antimicrobial activity, *M. smegmatis* mutants, DNA-damaging agents, structure–activity relationships

## Abstract

The emergence of drug resistance in pathogens leads to a loss of effectiveness of antimicrobials and complicates the treatment of bacterial infections. Quinoxaline 1,4-dioxides represent a prospective scaffold for search of new compounds with improved chemotherapeutic characteristics. Novel 2-acyl-3-trifluoromethylquinoxaline 1,4-dioxides with alteration of substituents at position 2 and 6 were synthesized via nucleophilic substitution with piperazine moiety and evaluated against a broad panel of bacteria and fungi by measuring their minimal inhibitory concentrations. Their mode of action was assessed by whole-genomic sequencing of spontaneous drug-resistant *Mycobacterium smegmatis* mutants, followed by comparative genomic analysis, and on an original pDualrep2 system. Most of the 2-acyl-3-trifluoromethylquinoxaline 1,4-dioxides showed high antibacterial properties against Gram-positive strains, including mycobacteria, and the introduction of a halogen atom in the position 6 of the quinoxaline ring further increased their activity, with **13c** being the most active compound. The mode of action studies confirmed the DNA-damaging nature of the obtained quinoxaline 1,4-dioxides, while drug-resistance may be provided by mutations in redox homeostasis genes, encoding enzymes potentially involved in the activation of the compounds. This study extends views about the antimicrobial and antifungal activities of the quinoxaline 1,4-dioxides and can potentially lead to the discovery of new antibacterial drugs.

## 1. Introduction

Bacterial infections have represented one of the most serious dangers to human health and animal husbandry throughout time. Currently there are over 12 classes of pathogenic bacteria, that cause various infectious diseases, which are considered as an increasing threat to human health due to their emerging resistance to most of antibacterial drugs, indicating the need for the development of novel agents for treating these drug-resistant bacteria [1]. Gram-negative bacteria, such as *Klebsiella pneumoniae* and *Escherichia coli,* can cause severe and even fatal infections, being a particular threat to people with weak or immature immunity, such as the newborn, the elderly, and patients with oncology or AIDS [2]. In addition, the emergence and spread of antibiotics-resistant pathogens limit the possibilities for treating infections. A particularly alarming tendency is the spread of multidrug- and totally drug-resistant bacteria (the so-called “superbugs”), which cause infections that cannot be treated with any of the existing antimicrobial drugs [3]. Tuberculosis (TB), caused by *Mycobacterium tuberculosis,* considered as the leading killer among bacterial infections caused by a single agent, is another example of a high spread of drug resistance. While the overall numbers of incidence and mortality are slowly decreasing throughout the years, the numbers of multidrug-resistant (MDR-TB, resistant to isoniazid and rifampicin) and extensively drug-resistant TB (XDR-TB, defined as MDR-TB with additional resistance to a fluoroquinolone and a second-line injectable drug) are constantly growing, representing a threat to global TB control [4]. Thus, search for alternatives to the currently used antimicrobial drugs, subjected to a loss of effectiveness due to the emergence of drug resistance, including targeted synthesis of new potential antibacterial agents is a hot topic direction in medicinal chemistry.

Quinoxaline 1,4-dioxides are an important class of heterocyclic compounds which are constantly in focus of medicinal chemists. The properties of quinoxaline 1,4-dioxides, which have a wide spectrum of biological activity, including antimicrobial, antiproliferative, antifungal, antiprotozoal, anti-inflammatory, and antioxidant activities, have been actively studied for more than five decades [5,6]. Due to their ability to undergo bioreductive activation, these derivatives are of great interest for a broad application in human and veterinary medicine.

The antibacterial activity of quinoxaline 1,4-dioxides has been known since the 1940s and they have also been used as growth stimulants which improve feed conversion efficiency in animal husbandry [7,8]. Carbadox (CARB, Figure 1, **1**) is an approved veterinary antibacterial agent used in the United States as a feed additive for the control and prevention of swine dysentery, helminthiasis and enteritis [5,9]. 2,3-Bis(hydroxymethyl)quinoxaline 1,4-dioxide (dioxidine, DIOX, **2**, Figure 1) is another derivative of quinoxaline 1,4-dioxide, which has found an application in the medical practice as effective antibacterial drug with a broad spectrum of antimicrobial activity [10]. Moreover, some quinoxaline 1,4-dioxides have demonstrated promising anti-yeast activity on *Candida albicans*, *C. glabrata*, *C. krusei*, *Saccharomyces cerevisiae* and the clinically important *C. parapsilosis* [11].

In the last two decades, several series of quinoxaline 1,4-dioxides have been evaluated and showed potent antibacterial activity, including activity against *M. tuberculosis* strains with MDR [12,13,14,15,16]. Previously, Monge et al. reported potential antitubercular quinoxaline 1,4-dioxide derivatives (Figure 1, compounds **3a**–**c**), which demonstrated activity against non-replicating mycobacteria responsible for the long-lasting courses of antitubercular therapy [17]. Nevertheless, the potential impact of the various substituents in quinoxaline nucleus on antimycobacterial activity requires further assessment. 

Thereby, quinoxaline 1,4-dioxide is a prospective scaffold for further development of new antimicrobial drugs with improved efficacy as well as lowered toxicity. Chemical modification of function groups attached to the quinoxaline scaffold is one of the methods for modulating its pharmacological properties. However, despite the great attention to bioactive quinoxaline 1,4-dioxides, the 6(7)-amino derivatives and their antimicrobial activity have not yet been studied. The trifluoromethyl group is one of the privileged structural fragments in medicinal chemistry which is used for modulating the pharmacological properties of molecules due to its unique characteristics such as molecular volume, lipophilicity, and the ability to form hydrogen bonds [18,19]. It has been previously revealed that the introduction of the trifluoromethyl group also significantly increased anti-mycobacterial activity of quinoxaline 1,4-dioxide derivatives [20]. Therefore, we assume that the introduction of a trifluoromethyl group into the structure of quinoxaline 1,4-dioxides can significantly increase both the binding to cellular targets and intracellular penetration of such derivatives. We report a series of water-soluble 2-acyl-3-trifluoromethylquinoxaline 1,4-dioxide derivatives with a variety of substituents at positions 2 and 6, which were synthesized and tested against a panel of pathogenic fungi, Gram-negative and Gram-positive bacteria, including *Mycobacterium* species.

## 2. Results and Discussion

### 2.1. Chemistry

It was previously shown that the halogen atoms in the phenyl ring of the quinoxaline 1,4-dioxide activated for nucleophilic substitution due to electron-withdrawing effects of heterocyclic moiety and attached substituents. This property may be efficiently used for a modulation of the properties of quinoxaline 1,4-dioxides [21]. Therefore, we applied this approach for the diversification of halogen-substituted 2-acyl-3-trifluoromethylquinoxaline 1,4-dioxides. Synthesis of several starting 3-trifluoromethylquinoxaline 1,4-dioxides (e.g., **5a–c**, **6a–c**) via the Beirut reaction has been described earlier [22,23,24,25]. According to this method, condensation of benzofuroxans **4a–d** [26,27] with trifluoromethylsubstituted 1,3-dicarbonyl compounds, which was performed in dry chloroform in the presence of triethylamine, led to a series of key 2-acyl-3-trifluoromethylquinoxaline 1,4-dioxides **5**–**6a–c**, **7a–d** and **8–11a–c**. Due to the tautomeric equilibrium in monosubstituted benzofuroxans, the two regioisomers of 6(7)-substituted quinoxaline 1,4-dioxides can be formed in the Beirut reaction, and their ratio strongly depends on the electronic effects of the substituents [28]. However, the reaction of the fluorine derivative **4a** with dicarbonyl compounds led exclusively to one quinoxaline derivative in all cases. Therefore, the position of the fluorine atom in the products obtained from fluorobenzofuroxan **4a** was fixedly examined. The analysis of ^13^C-NMR spectra was one of the reliable ways to determine the position of substituents in regioisomeric 6(7)-substituted quinoxaline 1,4-dioxides [28]. A comparison of the chemical shifts in the spectra of the paternal 2-propionyl-3-trifluoromethylquinoxaline 1,4-dioxide (**7d**) and its fluoro derivative **7a** revealed changes of the chemical shifts of bridging carbon atoms at the positions 9 and 10 of quinoxaline **7a** that corresponded to the described increments values (*I_C_*) [29] for the fluorine atom located at position 7 (Appendix A). It was therefore concluded that the assigned structure of the derivatives **5–11a** (Figure 1) showed a high regioselectivity of the Beirut reaction in the case of trifluoromethylsubstituted 1,3-dicarbonyl compounds with monosubstituted benzofuroxans.

The piperazine ring is a structural fragment of a lot of drugs actively applied in the medicine and veterinary [30]. Thus, piperazine moiety has important role in antimicrobial properties of fluoroquinolone antibiotics such as Ciprofloxacin and their analogues. This cyclic diamine and its derivatives were efficiently used as the *N*-nucleophiles for the diversification of the quinoxaline scaffolds and preparation of water-soluble derivatives with promising anticancer potencies [21,31]. However, attempts to carry out the nucleophilic substitution of the halogen atom on the quinoxaline core of **5–11a–c** by piperazine in DMF as described earlier [21] were ineffective due to the formation of a mixture of deoxygenated analogs as the main products of the reaction. A similar observation was made for several reactions of 3-trifluoromethylquinoxalines with different amines [32,33]. In addition, the isolation and purification of the target aminoderivatives were complicated by a higher water-solubility of their free bases as compared to the previously reported aminoquinoxaline 1,4 dioxides.

Nevertheless, the use of *N*-Boc-piperazine in this reaction was found to be more effective and reduced side products of the halogens substitution in quinoxaline ring of 3-trifluoromethylquinoxaline 1,4-dioxides **5–11**. Additionally, the protective group simplified the isolation and a chromatographic purification of formed Boc-derivatives **12–18** (Figure 2). The optimization of the reaction conditions including a variation of solvents, bases and the ratios of the reagent, allowed us for additionally increase the yields of Boc-derivative **12–18** by 20–40%. Thus, the treatment of derivatives **5–11a–c** with a threefold excess of *N*-Boc-piperazine in THF in the presence of the TEA at room temperature produced Boc-derivative of **12–18a**–**c** in good yields. The purified Boc-intermediates **12–18a–c** were transformed into the target hydrochlorides of amino-derivatives **12–18a–c** by the treatment with hydrogen chloride in ether (Figure 2). 

It should be noted that the treatment of the 6,7-dihalogeno-3-trifluoromethylquinoxaline 1,4-dioxides **5–11b–c** with *N*-Boc-piperazine resulted in the formation of predominantly 7-aminoderivatives **12–18b–c**. The position of the amino group in the quinoxaline ring of **12–18** was confirmed by analysis of the ^13^C-NMR spectra and the increment values for the chemical shifts. As mentioned above, the position of the substituents in the benzene ring of the quinoxaline has a significant effect on the chemical shifts of the signals of the pyrazine core in the ^13^C-NMR spectra. The comparison of the ^13^C-NMR spectra of 2-propionyl-3-trifluoromethylquinoxaline 1,4-dioxide (**7d**) and its amino derivatives **14a–c** showed that the largest changes in the chemical shift of carbon atoms were observed at positions 3 and 10 of the quinoxaline nucleus due to the conjugation with the amino group. The observed changes corresponded to the increments (*I_C_*) of the tertiary amino group in the ^13^C-NMR spectra of arenes [29,34] and well correlated to the data for previously described amino derivatives of quinoxaline 1,4-dioxides [21,31] (Appendix A). The structure of the obtained derivatives **14b–c** revealed that the halogen atoms at positions 6 and 7 of the starting compounds **5–11b–c** differed in the reactivity for the nucleophilic substitution. Thus, we assume, that the observed preferable substitution of the halogen atom at position 7 of 6,7-dihalogeno-3-trifluoromethylquinoxaline 1,4-dioxides **5–11b–c** is associated with a highest electron-withdrawal character of the trifluoromethyl group at position 3 in comparison with acyl groups at position 2 of the heterocycle that leads to formation of a more stable intermediate Mesenheimer complex. The structures of compounds **13b** and **13c** were additionally proven by 2D NMR techniques including HSQC and HMBC experiments (Appendix A): the presence of key four-bond correlations in the CIGAR-HMBC spectra between the signals of H-8 and C-2 observed for derivatives **13b** and **13c** confirmed the assigned structures of these compounds (Appendix A). It should be noted that the obtained results do not agree with the data of Zhang et al. [35], who described an analogue of compound **12b**, with the piperazine residue at the position 6 of heterocycle. However, the amine disposition was not justified in this work.

Along with the notable general instability of the obtained aminoderivatives of 3-trifluoromethylquinoxaline 1,4-dioxides **12–18**, compounds **15–18** with aroyl groups in the position 2 were the most unstable in this series under the reaction conditions, and during their isolation and purification. This partly explains the low yields of these derivatives. Moreover, in the case of 7-amino-2-aroyl-6-fluoro-3-trifluoromethylquinoxaline 1,4-dioxides **15–18b**, the fluorine atom at position 6 of the quinoxaline ring also retained the ability for nucleophilic substitution reactions. Thereby, the synthesis of these compounds alone to the target Boc-derivatives **15–18b** led to disubstitution byproducts **15–18d** (Figure 2) which were isolated as dihydrochlorides in 15–41% yields. The observed highest reactivity of the fluorine atom at position 6 of quinoxaline 1,4-dioxide **15–18b** in comparison to congeners **12–14b** in the substitution reaction could be explained by the biggest electron-withdrawing effect of the aroyl groups at position 2 of the quinoxaline ring of these derivatives.

We have earlier developed an original scheme for the synthesis of 7-aminoquinoxaline-2-carbonitrile 1,4-dioxides based on the Beirut reaction of 5-aminosubstituted benzofuroxans and benzoylacetonitrile, which proceeded with a good regioselectivity [31]. Considering this data, we tested the possibilities of this approach for the synthesis of congeners with 3-trifluoromethyl group, so a tentative heterocyclization of the previously described 5-(Boc-piperazin-1-yl)-6-chlorobenzofuroxan **19** [31] with 1,1,1-trifluorohexane-2,4-dione in THF in the presence of the triethylamine led to a formation of the derivative of quinoxaline 1,4-dioxide in a good yield (Figure 3). The obtained by this scheme compound had physicochemical and spectral characteristics identical to derivative **14c**, synthesized by the nucleophilic substitution of the chlorine atom of quinoxaline 1,4-dioxide **7c** (Figure 2). Thus, this alternative approach also led to 7-aminoderivatives 3-trifluoromethylquinoxaline 1,4-dioxides, however, the first one looking more preferable, since it opens up more opportunities for diversification of the scaffold on the final steps of the synthesis.

All synthesized compounds **12–18** were characterized by NMR spectra, HRMS and UV spectra; the HPLC-determined purity was ≥ 95% (Appendix A).

### 2.2. Biological Screening

The antibacterial activity of a novel 2-acyl-3-trifluoromethylquinoxaline 1,4-dioxides **12–18** was evaluated against a panel of bacterial strains, including Gram-positive (*S. aureus* ATCC 29213, *S. epidermidis* ATCC 14990, *E. faecalis* ATCC 29212, *M. smegmatis* mc^2^ 155, *M. tuberculosis* AlRa and *M. tuberculosis* UAlRv) and Gram-negative bacteria (*E. coli* ATCC 25922, *K. pneumoniae* 1951, *P. aeruginosa* ATCC 27853). The antifungal activity was evaluated against *C. albicans ATCC 10231* and *M. canis* B-200. The minimum inhibitory concentration (MIC) values of **12–18** and reference drugs are summarized in Table 1. DIOX was used as the main reference derivative of quinoxaline 1,4-dioxide used in the clinic, amphotericin B (AMPH) was used as a standard for the evaluation of antifungal activity, and rifampicin (RIF) and ciprofloxacin (CIP)–for the antibacterial activity. 

The structure and antibacterial activity of the 25 novel compounds are shown in Table 1. Eight derivatives out of the series (compounds **12c**, **13c**, **14c**, **15b**, **15c**, **16c**, **17c** and **18b**) had the highest activities against most of the tested microorganisms. It is worth noting that the MIC values of all the obtained compounds were noticeably lower than those of the reference drug DIOX. Furthermore, in some cases the obtained derivatives (for example, compounds **13c**, **15c**, **16c** and **18b**) have demonstrated similar antimicrobial potencies as compared to the reference antibiotics such as CIP, AMPH and RIF, but in general they had a lower activity against the tested cultures. Most of the compounds exhibited a promising activity on Gram-positive strains but were less effective on Gram-negative bacteria and fungi. However, the promising antibacterial activity on *E. coli ATCC 25922* and *K. pneumoniae 1951* was observed for the derivatives **12a**, **13a–c** and **14b** which have the MIC values in the range of 2–16 μg/mL. Also, the compounds **12c**, **15b** and **18b–d** have a comparable antifungal activity to AMPH (MIC = 4–16 and 0.75–2 μg/mL respectively) on *C. albicans ATCC 10231* and *M. canis* B-200. *P. aeruginsa ATCC 27853* was the least susceptible to the synthesized compounds. The lower activity observed on Gram-negative strains, with the exception of *E. coli ATCC* 25922, might be possibly caused by the lipopolysaccharide component of their cell wall, the lower permeability of the outer membrane, or some metabolic differences between these two phylogenetic groups, that provides the resistance to the compounds. At the present time, there is no evidence that could explain why some quinoxaline derivatives have differences in G+ and G− inhibition [36].

Analysis of the activity data presented in Table 1 reveals that the substitution of both halogen atoms, in positions 6 and 7 of the quinoxaline ring, by a cyclic diamine moiety led to a decrease of the antimicrobial activity of the derivatives **15–19d** against all tested strains. For instance, the derivatives of 6,7-diamino-3-trifluoromethylquinoxaline 1,4-dioxides **15–19d** displayed 2–8 times lower MIC values than their mono-amino analogs **15–19a**. At the same time, activity of the compound **18d** was similar to that of its mono- and halogen-substituted analogs **18a–c**. Regarding the type of the substituent at position 2 of quinoxaline, the derivatives with alkylcarbonyl- and alkoxy carbonyl groups (compounds **12–14**) slightly stand out in terms of MIC values compared to the compounds with aroyl and heteroaroyl groups in this position (derivatives **15–18** respectively). If considering aroyl groups at this position when the benzoyl is substituted by its heterocyclic bioisosteres (with furyl, thienyl) or naphthoyl moiety no principal changes in antimicrobial activity have been observed. Therefore, it can be concluded that the substituents on position 2 of the quinoxaline ring do not play an important role in antimicrobial activity. It is noteworthy that the most crucial structural requirement, is the presence of the halogen atom in position 7 of quinoxaline scaffold. In general, the highest antimicrobial activity in all the tested strains was shown by derivatives with a chlorine in 6 position of quinoxaline ring (compounds **12–18c**). It should be noted that the analogue of the synthesized derivatives, the reference drug DIOX, showed no activity in concentrations < 32 μg/mL against all tested microorganisms. Most of the derivatives demonstrated moderate to no antimycobacterial activity, however the compounds **12a**, **12c**, **13a**, **13c**, **14a**, **16c** and **18a** presented the same inhibitory effect as rifampicin (RIF) on *M. smegmatis mc^2^* 155 (MIC = 4 μg/mL), while the compounds **13b–c** and **15b–c** showed promising activity on *M. tuberculosis* strains with MIC values 5–10 μg/mL. The compound **13c** showed the highest activity against all tested microorganisms with the exception of *P. aeruginsa ATCC 27853* strain with MIC values between 0.25 and 10 μg/mL.

### 2.3. Mechanism of Action Determination

As it has been previously shown, that DIOX induces SOS-response in bacterial cells [37], the mechanism of antibacterial activity of novel 3-trifluoromethylquinoxaline 1,4-dioxides was analyzed by means of the pDualrep2 reported system [38]. In this reporter expression, *turboRFP* gene is under regulation of *sulA* promoter and upregulated in the presence of SOS-response inducers (topoisomerase inhibitors, DNA-intercalators etc), while *Katushka2S* is expressed in the presence of translation inhibitors, due to the stalling of the ribosomes at the modified TrpL leader peptide which leads to transcription antitermination. 

The obtained results, presented in Figure 3, show that most of the new derivatives of 3-trifluoromethylquinoxaline 1,4-dioxide **12–15** as well as reference drug DIOX strongly caused DNA damage and induced SOS response in *E. coli BW25513* strain, which corresponded with previous report of a DNA damaging mode of action of quinoxaline 1,4-dioxides induced by oxidative stress in pathogens [39]. Comparison of responses to DNA-damage between the DIOX and the new quinoxaline 1,4-dioxides showed that induction of the bacterial SOS-response for DIOX was more pronounced than the one for the compounds **12a**, **12c**, **13–15a–c**, **16b**, **16c**, **17a–c** and **18b**,**d**, however the majority of its analogues had a superior antibacterial activity. Additionally, as shown in Figure 3, the derivatives **13a–c** expressed both turboRFP and Katushka2S fluorescence proteins and may have a dual mechanism of action via DNA-damage and blocking of protein synthesis in bacterial cells. Interestingly, several compounds which have a noticeable antibacterial activity (e.g., **12b**, **15d**, **18d)** did not induce SOS-response or inhibition of protein synthesis, and so may act on bacteria via independent mechanisms and have an alternative bacterial target. 

Thereby, a primary analysis of structure-activity relationships and results of pDualrep2 assay revealed that the effect of 3-trifluoromethylquinoxaline 1,4-dioxide derivatives on bacterial cells is not limited to the action on DNA and can have a multitarget character. 

### 2.4. M. smegmatis Drug-Resistant Mutants and Their Analysis

The analysis of spontaneous drug-resistant mutants is a classic approach aimed at shedding additional light on the mechanisms of action and drug-resistance: mutations may occur in genes, encoding drug-targets, pro-drug activators, efflux-pumps, transcriptional regulators, proteins affecting cell wall permeability, etc., leading to drug resistance, and even cross-resistance with other drugs. We were able to obtain *M. smegmatis* mc^2^ 155 spontaneous mutants resistant to 4 × MIC of compounds **16b** and **15b** at a frequency of 3.6 × 10^−8^ and 5.0 × 10^−8^, respectively. We have selected 3 mutants, resistant to each compound (*M. smegmatis tfqR1, tfqR2* and *tfqR4* obtained on **16b**, and *M. smegmatis tfqR5, tfqR6* and *tfqR7* obtained on **15b**) for further analysis.

The mutants obtained on **16b** were cross-resistant to the majority of the tested compounds, with *M. smegmatis tfqR1* being the most resistant one, while those obtained on **15b** were only resistant to part of the compounds with just a slight increase in MIC level (Table 2).

The whole-genomic sequencing revealed that mutants *tfqR1, tfqR2* and *tfqR4* had 6–9 unique, non-synonymous mutations, and mutants *tfqR5, tfqR6* and *tfqR7–*2–3 unique, non-synonymous mutations each (Appendix A), supporting our data and the previous reports about quinoxaline 1,4-dioxides DNA-damaging properties [40,41].

One non-synonymous mutation was common for all the strains: a deletion of nine nucleotides (GCGCTGCTGC) in *MSMEG_4883*, resulting in a loss of three amino acids in the encoded AMP-dependent synthetase/ligase. We were able to construct the *M. smegmatis 4883c* strain by homologous recombination, harboring this mutation alone. However, this strain was susceptible to all the compounds, suggesting that this single mutation is not directly involved in resistance to the tested compounds, or might require one of the additional mutations to result in a drug-resistant phenotype. It is wortH-Noting, that the deleted 9 nucleotides (CGCTGCTGC) were repeated three times within the *MSMEG_4883* (Appendix A)*,* thus making the spontaneous occurrence of this mutation more likely to happen under the DNA-damaging effect of quinoxaline 1,4-dioxides.

We have also observed a mutation in *MSMEG_1380* in *M. smegmatis tfqR7,* encoding a TetR-family transcriptional repressor, which regulates the *mmpS5-mmpL5* operon expression. Mutations in *MSMEG_1380* lead to overexpression of *mmpS5-mmpL5* (*MSMEG_1381-MSMEG_1382*), resulting in enhanced efflux provided by the MmpS5-MmpL5 transporters [42]. The MmpS5-MmpL5 system is a multidrug resistance transporter, able to provide various mycobacterial species resistance to bedaquiline and clofazimine [43], azoles [44], thiacetazone derivatives [45], imidazo[1,2-b][1,2,4,5]tetrazines [46], and tryptanthrins [47]. The previously described recombinant *M. smegmatis atR9c* strain, with an incorporated frameshift mutation in *MSMEG_1380* [42] revealed to be sensitive to most of the tested compounds, though a 4-fold MIC increase was observed for compounds **13a**, **13b** and **18a**. Only a 2-fold increase in MIC was observed for **15b** on *M. smegmatis atr9c*, corresponding to the value on *M. smegmatis tfqR7*, which might be due to the differences of aeration in liquid and on solid media (used for MIC determination and mutants generation, respectively). Our data shows that the MmpS5-MmpL5 efflux might be structure-specific in regards to quinoxaline 1,4-dioxides, and act as a secondary mechanism of protection, providing a low level of resistance to some of the compounds.

Since quinoxaline 1,4-dioxides were shown to be redox-activated DNA-cleaving agents [41], some of the mutations in the resistant *M. smegmatis* mutants may lead to the disruption of the compounds activating enzymes. First of all, this refers to the non-synonymous single nucleotide polymorphisms (SNPs), observed in genes, encoding redox homeostasis enzymes, especially oxidases/oxidoreductases/dehydrogenases [39]. We observed nonsynonymous SNPs in MSMEG_0341, MSMEG_1497 and MSMEG_4323 (F420-dependent LLM class oxidoreductase, acyl-CoA dehydrogenase, and pyruvate dehydrogenase subunit E1, respectively) in *M. smegmatis tfqR2*, in MSMEG_4863 (LLM class flavin-dependent oxidoreductase) in *M. smegmatis tfqR4*, in MSMEG_0889 (succinate-semialdehyde dehydrogenase) in *M. smegmatis tfqR6*, and in MSMEG_6440 (monooxygenase, flavin-binding family protein) in *M. smegmatis tfqR7*. *M. smegmatis tfqR1* harbored an SNP in MSMEG_1914, encoding the alternative sigma-factor SigH. Its homolog in *M. tuberculosis* has been shown to be induced by heat, oxidative and nitric oxide stresses, upregulating a plethora of genes, including several oxidoreductases [47], thus the disruption of its function may lead to the downregulation of several genes, potentially involved in activation of quinoxaline 1,4-dioxides.

Mutations in MSMEG_4189 and MSMEG_6392, observed in *M. smegmatis tfqR5* may probably have a direct or indirect protective effect on bacterial cells. MSMEG_4189 encodes the cysteinyl-tRNA synthetase, which is involved in mycothiol biosynthesis pathway in mycobacteria. Mycothiol acts as an electron acceptor/donor and serves as a cofactor in detoxification reactions for alkylating agents, free radicals and xenobiotics, maintaining the intracellular redox homeostasis [48]. MSMEG_6392 encodes the polyketide synthase Pks13, an enzyme which catalyzes the condensation reaction, which produces α-alkyl-β-ketoacids, direct precursors of mycolic acids, thus being involved in cell wall biosynthesis [49]. The mutation in MSMEG_6392 could thus alter the permeability of the mycobacterial cell wall.

The number of unique mutations within a single strain correlated with the level of resistance, suggesting that their combination has a synergistic effect. However, the mutation in SigH might have the highest stand-alone effect for *M. smegmatis tfqR1* due the downregulation of several oxidoreductases by its disruption, while the MSMEG_4863-encoded oxidoreductase, could have a higher impact in the activation of quinoxaline 1,4-dioxides alone, as compared to other redox-proteins, as its mutation led to a higher level of resistance, as compared to those in MSMEG_0889 and MSMEG_6440.

## 3. Materials and Methods

### 3.1. Synthesis

#### 3.1.1. Materials and General Methods

NMR spectra of all synthesized compounds were recorded on a VXR-400 instrument (Varian, 3100 Hansen Way, Palo Alto, CA, USA) at 400 MHz (^1^H-NMR) and 100 MHz (^13^C-NMR). Chemical shifts were measured in DMSO-*d_6_* using TMS as an internal standard. Analytical TLC was performed on silica gel F254 plates (Merck KGaA, 64271 Darmstadt, Germany) and column chromatography on Merck 60 Silica Gel. Melting points were determined on an SMP-20 apparatus (Büchi, CH-9230 Flawil 1, Flawil, Switzerland) and are uncorrected. High-resolution mass spectra were recorded by electron spray ionization on a micro OTOF-QII instrument (Bruker Daltonics, Bremen, Gernany). UV spectra were recorded on a U2000 spectrophotometer (Unico, 182 Ridge Road, Dayton, USAIR spectra were recorded on iS10 Fourier transform IR spectrometer (Nicolet, Madison, WI, USA). HPLC was performed using a Class-VP V6.12SP1 system (Shimadzu, Canby, Oregon, USA) equipped with a GraseSmart RP-18, 6 × 250 mm column (Eka Chemicals AB, Bohus, Sweden). Eluents: A, H_3_PO_4_ (0.01 M); B, MeCN. All solutions were evaporated at reduced pressure on a Büchi-R200 rotary evaporator at a temperature below 50 °C. All products were dried under vacuum at room temperature. Unless specified otherwise all solvents, chemicals, and reagents were obtained from Sigma-Aldrich (3050 Spruce Street, St. Louis, USA) and used without purification. The purity of compounds **12–18** was >95% as determined by HPLC analysis. All products were dried under vacuum at room temperature. The corresponding benzofuroxans **4a–d** were prepared as described [50]. 

#### 3.1.2. General Procedure for Synthesis of **5–6a–c**, **7a–d** and **8–11a–c**

The compounds **5–6a–c**, **7a–d** and **8–11a–c** were previously synthesized in [22,23,24,25]. The preparation of the compounds **5–6a–c**, **7a–d** and **8–11a–c** was carried out by the Beirut reaction (Figure 1). The appropriate benzofuroxanes **4a**–**d** (2.4 mmol) were obtained by previously described methods [50] and appropriate trifluoromethyl-substituted 1,3-dicarbonyl compounds (9.6 mmol) were dissolved in the mixture of dry chloroform (10 mL) and triethylamine (1 mL). When the reaction was finished, the solvent was evaporated and resulted yellow solid or brown oil was precipitated from dichloromethane-hexane mixture (1:3). The products were purified by flash chromatography on a silica gel using eluting solvent (toluene-ethyl acetate mixture, 5:1). 

#### 3.1.3. 2-Acetyl-7-fluoro-3-trifluoromethylquinoxaline 1,4-Dioxide (**6a**)

This compound was obtained in 72% yield from benzofuroxane **4a** and 1,1,1-trifluoropentane-2,4-dione following the general procedure, mp 153–155 °C. ^1^H-NMR (400 MHz, DMSO-*d*_6_) δ 8.60 (1H, dd, *J^1^* = 9.8, *J*^2^ = 5.1, H-8); 8.24 (1H, dd, *J*^1^ = 9.0, *J^2^* = 2.7, H-5); 8.01 (1H, td, *J*^1^ = 9.0, *J*^2^ = 2.7, H-6); 2.61 (3H, s, CH_3_). ^13^C-NMR (100 MHz, DMSO-*d*_6_) δ 191.4 (CO); 164.8 (d, *J* = 256.9, 7-CF); 139.8 (d, *J* = 11.5, 9-C); 139.2 (2-C); 135.8 (10-C); 127.7 (q, *J* = 33.7, 3-C); 123.7 (d, *J* = 9.9, 5-CH); 122.9 (d, *J* = 26.1, 6-CH); 119.1 (q, *J* = 272.9, CF_3_); 105.3 (d, *J* = 28.4, 8-CH). HRMS (ESI) calculated for C_11_H_7_F_4_N_2_O_3_^+^ [M+H]^+^ 291.0387, found 291.0414.

#### 3.1.4. 7-Fluoro-2-propionyl-3-trifluoromethylquinoxaline 1,4-Dioxide (**7a**)

This compound was obtained in 83% yield from benzofuroxane **4a** and 1,1,1-trifluorohexane-2,4-dione following the general procedure, mp 159–160 °C. ^1^H-NMR (400 MHz, DMSO-*d*_6_) δ 8.60 (1H, dd, *J^1^* = 9.9, *J^2^* = 4.8, H-8); 8.24 (1H, br. d, *J* = 7.8, H-5); 8.01 (1H, td, *J*^1^ = 7.8, *J*^2^ = 2.5, H-6); 2.92 (2H, q, *J* = 7.2, **CH_2_**CH_3_); 1.16 (3H, t, *J* = 7.2, CH_2_**CH_3_**). ^13^C-NMR (100 MHz, DMSO-*d*_6_) δ 194.6 (CO) 164.8 (d, *J* = 256.3, 7-CF); 139.8 (d, *J* = 12.2, 9-C); 139.1 (2-C); 135.9 (10-C); 127.9 (q, *J* = 36.6, 3-C); 123.7 (d, *J* = 9.9, 5-CH); 122.8 (d, *J* = 25.9, 6-CH); 119.1 (d, *J* = 273.9, CF_3_); 105.3 (d, *J* = 28.2, 8-CH). HRMS (ESI) calculated for C_12_H_9_F_4_N_2_O_3_ [M+H]^+^ 305.0544, found 305.0507.

#### 3.1.5. 2-Propionyl-3-trifluoromethylquinoxaline 1,4-Dioxide (**7d**)

This compound was obtained in 59% yield from benzofuroxane **4d** and 1,1,1-trifluorohexane-2,4-dione following the general procedure, mp 178–179 °C (139–140 °C, [23]). ^1^H-NMR (400 MHz, DMSO-*d*_6_) δ 8.52 (1H, dd, *J*^1^ = 8.2, *J*^2^ = 1.6, H-8); 8.46 (1H, dd, *J*^1^ = 8.2, *J*^2^ = 1.6, H-5); 8.12 (1H, td, *J*^1^ = 7.0, *J*^2^ = 1.6, H-7); 8.08 (1H, td, *J*^1^ = 7.0, *J*^2^ = 1.6, H-6); 2.92 (2H, q, *J* = 7.0, CH_3_**CH_2_**CO); 1.16 (3H, t, *J* = 7.0, **CH_3_**CH_2_CO). ^13^C-NMR (100 MHz, DMSO-*d*_6_) δ 194.9 (CO) 138.4 (9-C); 138.4 (2-C); 138.3 (10-C); 134.2 (6-CH); 133.4 (7-CH); 128.2 (q, *J* = 32.0, 3-C); 120.1 (8-CH); 119.7 (5-CH); 119.2 (q, *J* = 273.8, CF_3_); 35.2 (CH_3_**CH_2_**CO); 7.1 (**CH_3_**CH_2_CO). 

#### 3.1.6. 2-Ethoxycarbonyl-7-(piperazin-1-yl)-3-trifluoromethylquinoxaline 1,4-Dioxide Hydrochloride (**12a**)

To a solution of 3-carboethoxy-7-fluoro-3-trifluoromethylquinoxaline 1,4-dioxide **5a** (150 mg, 0.47 mmol) in tetrahydrofuran (10 mL) *N*-Boc-piperazine (260 mg, 1.4 mmol) and triethylamine (100 μL) were added and the mixture was stirred at room temperature for 5 h. After the reaction was completed (determined by TLC), the reaction mixture was evaporated, the residue was purified by column chromatography on a silica gel in chloroform-acetone (10:1). The residue was crystallized from dichloromethane-*n*-hexane mixture. Obtained Boc-**12a** was dissolved in tetrahydrofuran (3 mL) and the 3M solution of HCl in diethyl ether (2 mL) was added. The reaction mixture was stirred at room temperature for 3h and evaporated. The crystals were dissolved in water (1 mL), filtered, the product was precipitated with methanol-ether mixture (1:3). The formed precipitate was collected by filtration, washed with mixture of methanol-diethyl ether (1:5), n-hexane and dried. The yield of hydrochloride of **12a** was 84 mg (46%) as an orange powder, mp 200–202 °C (dec.). HPLC (LW = 335 nm, gradient B 20/60% (45 min)) t_R_ = 12.9 min, purity 98.3%. λ_max._, EtOH: 272, 302, 337, 382 nm. ^1^H-NMR (400 MHz, DMSO-*d*_6_) δ 9.64 (2H, br. s, NH_2_^+^); 8.27 (1H, d, *J* = 9.0, H-5); 7.84 (1H, d, *J* = 9.0, H-6); 7.48 (1H, s, H-8); 4.45 (2H, q, *J* = 7.0, O**CH_2_**CH_3_); 3.82 (4H, m, CH_2_); 3.27 (4H, m, CH_2_); 1.32 (3H, t, *J* = 7.0, OCH_2_**CH_3_**). ^13^C-NMR (100 MHz, DMSO-*d*_6_) δ 158.1 (CO); 152.9 (7-C); 139.6 (9-C); 133.6 (3-C); 131.5 (10-C); 124.3 (q, *J* = 33.6, 3-C); 122.7 (6-CH); 121.4 (5-CH); 119.3 (q, *J* = 272.4, CF_3_); 97.8 (8-CH); 63.5 (O**CH_2_**CH_3_); 43.6 (2 × CH_2_); 42.7 (2 × CH_2_); 13.6 (OCH_2_**CH_3_**). HRMS (ESI) calculated for C_16_H_18_F_3_N_4_O_4_^+^ [M+H]^+^ 387.1275, found 387.1250.

#### 3.1.7. 2-Ethoxycarbonyl-6-fluoro-7-(piperazin-1-yl)-3-trifluoromethylquinoxaline 1,4-Dioxide Hydrochloride (**12b**)

This compound was prepared from **5b** and *N*-Boc-piperazine as described for **12a**. A yellow powder, yield 42%, mp 208–209 °C (dec.). HPLC (LW = 340 nm, gradient B 20/80% (45 min)) t_R_ = 11.3 min, purity 96.5%. λ_max._, EtOH: 213, 272, 342, 382 nm. ^1^H-NMR (400 MHz, DMSO-*d*_6_) δ 9.69 (2H, s, NH_2_^+^); 8.24 (1H, d, *J* = 12.9, H-5); 7.73 (1H, d, *J* = 8.2, H-8); 4.47 (2H, q, *J* = 7.0, O**CH_2_**CH_3_); 3.62 (2H, br. m, CH_2_); 3.29 (br. m, 4H, CH_2_); 1.32 (3H, t, *J* = 7.0, OCH_2_**CH_3_**). ^13^C-NMR (100 MHz, DMSO-*d*_6_) δ 157.9 (CO) 157.9 (d, *J* = 259.2, 7-CF); 144.9 (d, *J* = 10.7, 6-C); 136.3 (2-C); 133.9 (d, *J* = 11.5, 9-C); 133.4 (10-C); 126.6 (d, *J* = 33.7, 3-C); 119.0 (q, *J* = 273.8, CF_3_); 107.0 (d, *J* = 29.1, 8-CH); 105.9 (5-CH); 63.7 (O**CH_2_**CH_3_); 46.3 (2 × CH_2_); 42.3 (2 × CH_2_); 13.7 (OCH_2_**CH_3_**). HRMS (ESI) calculated for C_16_H_17_F_4_N_4_O_4_^+^ [M+H]^+^ 405.1180, found 405.1233.

#### 3.1.8. 2-Ethoxycarbonyl-6-chloro-7-(piperazin-1-yl)-3-trifluoromethylquinoxaline 1,4-Dioxide Hydrochloride (**12c**)

This compound was prepared from **5c** and *N*-Boc-piperazine as described for **12a**. A yellow powder, yield 83%, mp 253–254 °C (dec.). HPLC (LW = 280 nm, gradient B 20/80% (455 min)) t_R_ = 13.9 min, purity 96.6%. λ_max._, EtOH: 235, 267, 282, 345, 393 nm. ^1^H-NMR (400 MHz, DMSO-*d*_6_) δ 8.97 (2H, br. s, NH_2_^+^); 8.50 (1H, s, H-8); 7.86 (1H, s, H-5); 4.47 (2H, q, *J* = 7.0, O**CH_2_**CH_3_); 4.31 (4H, br. m, CH_2_); 3.46 (4H, br. m, CH_2_); 1.32 (3H, q, *J* = 7.0, OCH_2_**CH_3_**). ^13^C-NMR (100 MHz, DMSO-*d*_6_) δ 157.8 (CO); 152.8 (7-C); 138.0 (9-C); 134.6 (3-C); 134.4 (10-C); 133.8 (6-CCl); 127.3 (d, *J* = 33.7, 3-C); 121.9 (5-CH); 118.9 (q, *J* = 272.9, CF_3_); 108.6 (8-CH); 63.8 (O**CH_2_**CH_3_); 47.6 (2 × CH_2_); 42.9 (2 × CH_2_); 13.7 (OCH_2_**CH_3_**). HRMS (ESI) calculated for C_16_H_17_ClF_3_N_4_O_4_^+^ [M+H]^+^ 421.0885, found 421.0883.

#### 3.1.9. 2-Acetyl-7-(piperazin-1-yl)-3-trifluoromethylquinoxaline 1,4-Dioxide Hydrochloride (**13a**)

This compound was prepared from **6a** and *N*-Boc-piperazine as described for **12a**. A yellow powder, yield 81%, mp 260–261 °C (dec.). HPLC (LW = 336 nm, gradient B 10/90% (45 min)) t_R_ = 12.3 min, purity 94.6%. λ_max._, EtOH: 216, 271, 301, 336, 381 nm. ^1^H-NMR (400 MHz, DMSO-*d*_6_) δ 9.73 (2H, br. s, NH_2_^+^); 8.30 (1H, d, *J* = 9.6, H-5); 7.86 (1H, d, *J* = 9.6, H-6); 7.51 (1H, s, H-8); 3.83 (4H, br. m, CH_2_); 3.24 (4H, br. m, CH_2_); 2.58 (3H, s, CH_3_). ^13^C-NMR (100 MHz, DMSO-*d*_6_) δ 191.9 (CO); 152.9 (7-C); 139.6 (9-C); 138.8 (10-C); 131.4 (d, *J* = 9.2, 2-C); 124.2 (d, *J* = 33.7, 3-C); 122.5 (5-CH); 121.4 (6-CH); 119.5 (q, *J* = 272.9, CF_3_); 97.6 (8-CH); 43.7 (2 × CH_2_); 42.1 (2 × CH_2_); 29.3 (CH_3_). HRMS (ESI) calculated for C_15_H_16_F_3_N_4_O_3_^+^ [M+H]^+^ 357.1169, found 357.1197.

#### 3.1.10. 2-Acetyl-6-fluoro-7-(piperazin-1-yl)-3-trifluoromethylquinoxaline 1,4-Dioxide Hydrochloride (**13b**)

This compound was prepared from **6b** and *N*-Boc-piperazine as described for **12a**. A yellow powder, yield 63%, mp 255–257 °C (dec.). HPLC (LW = 335 nm, gradient B 20/80% (45 min)) t_R_ = 9.6 min, purity 96.0%. λ_max._, EtOH: 216, 274, 334, 381, 423 nm. ^1^H-NMR (400 MHz, DMSO-*d*_6_) δ 9.79 (2H, s, NH_2_^+^); 8.24 (1H, d, *J* = 12.9, H-5); 7.74 (1H, d, *J* = 7.8, H-8); 3.65 (4H, br. m, CH_2_); 3.29 (4H, br. m, CH_2_); 2.59 (3H, s, CH_3_). ^13^C-NMR (100 MHz, DMSO-*d*_6_) δ 191.5 (CO); 157.5 (d, *J* = 259.2, 6-CF); 144.9 (d, *J* = 10.7, 6-C); 138.5 (9-C); 136.2 (2-C); 133.5 (d, *J* = 11.5, 10-C); 126.4 (d, *J* = 34.5, 3-C); 119.1 (q, *J* = 273.8, CF_3_); 106.8 (d, *J* = 29.1, 5-CH); 105.7 (8-CH); 46.3 (2 × CH_2_); 42.2 (2 × CH_2_); 29.3 (CH_3_). HRMS (ESI) calculated for C_15_H_15_F_4_N_4_O_3_^+^ [M+H]^+^ 375.1075, found 375.1167.

#### 3.1.11. 2-Acetyl-7-chloro-6-(piperazin-1-yl)-3-trifluoromethylquinoxaline 1,4-Dioxide Hydrochloride (**13c**)

This compound was prepared from **6c** and *N*-Boc-piperazine as described for **12a**. A yellow powder, yield 78%, mp > 260 °C (dec.). HPLC (LW = 340 nm, gradient B 10/90% (45 min)) t_R_ = 14.0 min, purity 97.4%. λ_max._, EtOH: 222, 244, 282, 342, 391 nm. ^1^H-NMR (400 MHz, DMSO-*d*_6_) δ 9.82 (2H, br. s, NH_2_^+^); 8.52 (1H, s, H-5); 7.86 (1H, s, H-8); 3.53–3.51 (4H, m, CH_2_); 3.31–3.30 (4H, m, CH_2_); 2.60 (3H, s, CH_3_).^13^C-NMR (100 MHz, DMSO-*d*_6_) δ 191.4 (CO); 152.8 (7-C); 138.9 (9-C); 137.9 (10-C); 134.2 (6-CCl); 134.1 (2-C); 127.1 (d, *J* = 33.7, 3-C); 121.7 (5-CH); 119.1 (q, *J* = 273.7, CF_3_); 108.2 (8-CH); 47.4 (2 × CH_2_); 42.5 (2 × CH_2_); 29.2 (CH_3_). HRMS (ESI) calculated for C_15_H_15_ClF_3_N_4_O_3_^+^ [M+H]^+^ 391.0779, found 391.0797.

#### 3.1.12. 7-(Piperazin-1-yl)-2-propionyl-3-trifluoromethylquinoxaline 1,4-Dioxide Hydrochloride (**14a**)

This compound was prepared from **7a** and *N*-Boc-piperazine as described for **12a**. A yellow powder, yield 76%, mp 180–182 °C (dec.). HPLC (LW = 300 nm, gradient B 20/80% (45 min)) t_R_ = 16.2 min, purity 95.5%. λ_max._, EtOH: 272, 302, 337, 371, 382 nm. ^1^H-NMR (400 MHz, DMSO-*d*_6_) δ 8.99 (2H, br. s, NH_2_^+^); 8.31 (1H, d, *J* = 8.0, H-5); 7.86 (1H, d, *J* = 8.0, H-6); 7.52 (1H, s, H-8); 3.79 (4H, br. m, CH_2_); 3.45 (4H, br. m, CH_2_); 2.90 (2H, br. m, CO**CH_2_**CH_3_); 1.14 (3H, br. m, COCH_2_**CH_3_**). ^13^C-NMR (100 MHz, DMSO-*d*_6_) δ 195.2 (CO); 152.9 (7-C); 139.6 (9-C); 138.7 (2-C); 131.4 (10-C); 124.4 (d, *J* = 33.6, 3-C); 122.6 (6-CH); 121.4 (5-CH); 119.5 (q, *J* = 272.4, CF_3_); 97.6 (8-CH); 43.8 (2 × CH_2_); 42.4 (2 × CH_2_); 35.1 (**CH_2_**CH_3_); 7.2 (CH_2_**CH_3_**). HRMS (ESI) calculated for C_11_H_4_F_7_N_2_O_3_^+^ [M+H]^+^ 371.1326, found 371.1338.

#### 3.1.13. 6-Fluoro-7-(piperazin-1-yl)-2-propionyl-3-trifluoromethylquinoxaline 1,4-Dioxide Hydrochloride (**14b**)

This compound was prepared from **7b** and *N*-Boc-piperazine as described for **12a**. A yellow powder, yield 58%, mp > 260 °C (dec.). HPLC (LW = 335 nm, gradient B 10/90% (45 min)) t_R_ = 14.5 min, purity 95.4%. λ_max._, EtOH: 216, 274, 335, 388 nm. ^1^H-NMR (400 MHz, DMSO-*d*_6_) δ 9.77 (2H, br. s, NH_2_^+^); 8.24 (1H, d, *J* = 12.9, H-5); 7.74 (1H, d, *J* = 8.2, H-8); 3.64 (4H, br. m, CH_2_); 3.37–3.35 (4H, m, CH_2_); 2.90 (2H, q, *J* = 7.3, CO**CH_2_**CH_3_); 1.14 (3H, t, *J* = 7.3, COCH_2_**CH_3_**). ^13^C-NMR (100 MHz, DMSO-*d*_6_) δ 194.7 (CO); 157.5 (d, *J* = 259.2, 6-CF); 144.8 (d, *J* = 11.5, 7-C); 138.4 (9-C); 136.2 (2-C); 133.5 (d, *J* = 34.5, 10-C); 126.5 (d, *J* = 34.5, 3-C); 118.3 (q, *J* = 273.8, CF_3_); 106.8 (d, *J* = 29.1, 5-CH); 105.6 (8-CH); 46.2 (2 × CH_2_); 42.2 (2 × CH_2_); 35.1 (**CH_2_**CH_3_); 7.1 (CH_2_**CH_3_**). HRMS (ESI) calculated for C_16_H_16_F_4_N_4_O_3_ [M+H]^+^ 389.1231, found 389.1221.

#### 3.1.14. 6-Chloro-7-(piperazin-1-yl)-2-propionyl-3-trifluoromethylquinoxaline 1,4-Dioxide Hydrochloride (**14c**)

This compound was prepared from **7c** and *N*-Boc-piperazine as described for **12a**. A yellow powder, yield 45%, mp 257–258 °C (dec.). HPLC (LW = 300 nm, gradient B 20/80% (45 min)) t_R_ = 15.8 min, purity 97.3. λ_max._, EtOH: 223, 244, 342, 392 nm. ^1^H-NMR (400 MHz, DMSO-*d*_6_) δ 9.03 (2H, br. s, NH_2_^+^); 8.51 (1H, s, H-8); 7.85 (1H, s, H-5); 3.47 (4H, br. m, CH_2_); 3.35 (4H, br. m, CH_2_); 2.90 (2H, q, *J* = 6.8, CO**CH_2_**CH_3_); 1.13 (3H, t, *J* = 6.8, COCH_2_**CH_3_**).^13^C-NMR (100 MHz, DMSO-*d*_6_) δ 194.8 (CO); 152.8 (7-C); 138.9 (9-C); 138.0 (2-C); 134.3 (10-C); 134.3 (6-CCl); 127.3 (d, *J* = 34.3, 3-C); 121.6 (5-CH); 119.2 (q, *J* = 273.2, CF_3_); 108.4 (8-CH); 47.6 (2 × CH_2_); 42.9 (2 × CH_2_); 30.7 (**CH_2_**CH_3_); 7.2 (CH_2_**CH_3_**). HRMS (ESI) calculated for C_16_H_17_ClF_3_N_4_O_3_^+^ [M+H]^+^ 405.0936, found 405.0947.

#### 3.1.15. 2-Benzoyl-7-(piperazin-1-yl)-3-trifluoromethylquinoxaline 1,4-Dioxide Hydrochloride (**15a**)

This compound was prepared from **8a** and *N*-Boc-piperazine as described for **12a**. An orange powder, yield 74%, mp 234–236 °C (dec.). HPLC (LW = 340 nm, gradient B 10/90% (45 min)) t_R_ = 8.79 min, purity 97.7%. λ_max._, EtOH: 206, 275, 340, 383 nm. ^1^H-NMR (400 MHz, DMSO-*d*_6_) δ 9.65 (2H, br. s, NH_2_^+^); 8.38 (1H, d, *J* = 9.8, H-5); 8.09 (2H, d, *J* = 7.8, C_6_H_5_); 7.89 (1H, d, *J* = 9.8, H-6); 7.75 (1H, t, *J* = 7.8, C_6_H_5_); 7.58 (2H, t, *J* = 7.8, C_6_H_5_); 7.49 (1H, s, H-8); 3.82 (4H, br. m, CH_2_); 3.25 (4H, br. m, CH_2_). ^13^C-NMR (100 MHz, DMSO-*d*_6_) δ 184.6 (CO); 152.9 (7-C); 139.9 (9-C); 137.4 (1′-C); 135.2 (4′-CH); 134.1 (2-C); 132.0 (10-C); 129.3 (2 × 2′-CH); 129.2 (2 × 3′-CH); 124.7 (d, *J* = 46.7, 3-C); 122.6 (6-CH); 121.5 (5-CH); 119.5 (q, *J* = 272.9, CF_3_); 43.7 (2 × CH_2_); 42.1 (2 × CH_2_). HRMS (ESI) calculated for C_20_H_18_F_3_N_4_O_3_^+^ [M+H]^+^ 419.1326, found 419.1327.

#### 3.1.16. 2-Benzoyl-6-fluoro-7-(piperazin-1-yl)-3-trifluoromethylquinoxaline 1,4-Dioxide Hydrochloride (**15b**)

This compound was prepared from **8b** and *N*-Boc-piperazine as described for **12a**. R_f_ = 0.47 (chloroform-acetone, 5:1). A yellow powder, yield 52%, mp 215–216 °C (dec.). HPLC (LW = 270 nm, gradient B 20/80% (45 min)) t_R_ = 14.7 min, purity 95.1%. λ_max._, EtOH: 271, 338, 391 nm. ^1^H-NMR (400 MHz, DMSO-*d*_6_) δ 9.71 (2H, br. s, NH_2_^+^); 8.32 (1H, d, *J* = 12.5, H-5); 8.10 (2H, d, *J* = 8.2, C_6_H_5_); 7.76 (1H, t, *J* = 7.4, C_6_H_5_); 7.71 (1H, d, *J* = 8.2, H-8); 7.59 (2H, t, *J* = 7.4, C_6_H_5_); 3.62 (4H, br. m, CH_2_); 3.29 (4H, br. m, CH_2_). ^13^C-NMR (100 MHz, DMSO-*d*_6_) δ 184.3 (CO); 157.6 (d, *J* = 259.2, 6-CF); 144.9 (d, *J* = 11.5, 7-C); 137.3 (9-C); 136.7 (2-C); 135.3 (4′-CH); 134.3 (d, *J* = 11.5, 10-C); 134.0 (1′-C); 129.4 (2 × 2′-CH); 129.3 (2 × 3′-CH); 127.6 (d, *J* = 32.9, 3-C); 119.3 (q, *J* = 272.9, CF_3_); 107.0 (d, *J* = 28.4, 5-CH); 105.9 (d, *J* = 3.1, 8-CH); 46.3 (2 × CH_2_); 42.3 (2 × CH_2_). HRMS (ESI) calculated for C_20_H_17_F_4_N_4_O_3_^+^ [M+H]^+^ 437.1231, found 437.1253.

#### 3.1.17. 2-Benzoyl-6-chloro-7-(piperazin-1-yl)-3-trifluoromethylquinoxaline 1,4-Dioxide Hydrochloride (**15c**)

This compound was prepared from **8c** and *N*-Boc-piperazine as described for **12a**. A yellow powder, yield 84%, mp 243–244 °C(dec.). HPLC (LW = 345 nm, gradient B 10/90% (45 min)) t_R_ = 17.5 min, purity 96.7%. λ_max._, EtOH: 258, 284, 345, 394 nm. ^1^H-NMR (400 MHz, DMSO-*d*_6_) δ 9.88 (2H, br. s, NH_2_^+^); 8.57 (1H, s, H-8); 8.10 (2H, d, *J* = 7.8, C_6_H_5_); 7.82 (1H, s, H-5); 7.74 (1H, m, C_6_H_5_); 7.59 (2H, t, *J* = 7.8, C_6_H_5_); 3.51 (4H, br. m, CH_2_); 3.29 (4H, br. m, CH_2_). ^13^C-NMR (100 MHz, DMSO-*d*_6_) δ 184.2 (CO); 152.7 (7-C); 139.9 (9-C); 138.3 (10-C); 137.7 (1′-C); 135.4 (4′-CH); 134.9 (6-CCl); 134.0 (d, *J* = 24.5, 3-C); 131.8 (2-C); 129.3 (2 × 2′-CH); 129.2 (2 × 3′-CH); 121.9 (5-CH); 108.3 (8-CH); 119.1 (q, *J* = 274.5, CF_3_); 47.4 (2 × CH_2_); 42.6 (2 × CH_2_). HRMS (ESI) calculated for C_20_H_17_ClF_3_N_4_O_3_^+^ [M+H]^+^ 453.0936, found 453.0918.

#### 3.1.18. 2-Benzoyl-6,7-di(piperazin-1-yl)-3-trifluoromethylquinoxaline 1,4-Dioxide Hydrochloride (**15d**)

This compound was isolated as a minor component in the synthesis of the derivative **15b** described in 3.1.16. R_f_ = 0.29 (chloroform-acetone, 5:1). An orange powder, yield 20%, mp >260 °C (dec.). HPLC (LW = 335 nm, gradient B 10/90% (45 min)) t_R_ = 12.80 min, purity 97.9%. λ_max._, EtOH: 258, 338, 441 nm. ^1^H-NMR (400 MHz, DMSO-*d*_6_) δ 7.99 (2H, d, *J* = 7.8, C_6_H_5_); 7.81 (1H, s, H-8); 7.73 (2H, t, *J* = 7.8, C_6_H_5_); 7.66 (1H, s, H-5); 7.56 (1H, t, *J* = 7.8, C_6_H_5_); 3.52 (8H, br. m, CH_2_); 3.35 (8H, br. m, CH_2_). ^13^C-NMR (100 MHz, DMSO-*d*_6_) δ 185.1 (CO) 150.6 (7-C); 149.8 (6-C); 144.9 (9-C); 136.9 (2-C); 136.0 (1′-C); 135.7 (4′-CH); 134.5 (10-C); 130.1 (2 × 2′-CH); 129.7 (2 × 3′-CH); 127.9 (d, *J* = 33.7, 3-C); 107.9 (8-CH); 107.1 (5-CH); 49.7 (2 × CH_2_); 47.7 (2 × CH_2_); 46.3 (2 × CH_2_); 46.1 (2 × CH_2_); 43.1 (2 × CH_2_); 43.1 (2 × CH_2_). HRMS (ESI) calculated for C_24_H_26_F_3_N_6_O_3_^+^ [M+H]^+^ 503.2013, found 503.2008.

#### 3.1.19. 2-(Furan-2-carbonyl)-7-(piperazin-1-yl)-3-trifluoromethylquinoxaline 1,4-Dioxide Hydrochloride (**16a**)

This compound was prepared from **9a** and *N*-Boc-piperazine as described for **12a**. An orange powder, yield 71%, mp 220–221 °C (dec.). HPLC (LW = 340 nm, gradient B 10/90% (45 min)) t_R_ = 11.0 min, purity 97.9%. λ_max._, EtOH: 217, 248, 275, 297, 341, 384, 460 nm. ^1^H-NMR (400 MHz, DMSO-*d*_6_) δ 9.60 (2H, br. s, NH_2_^+^); 8.37 (1H, d, *J* = 8.9, H-5); 8.20 (1H, br. m, C_4_H_3_O); 7.89 (1H, d, *J* = 8.9, H-6); 7.85 (1H, br. m, C_4_H_3_O); 7.50 (1H, s, H-8); 6.82 (1H, br. m, C_4_H_3_O); 3.81 (4H, br. m, CH_2_); 3.24 (4H, m, CH_2_). ^13^C-NMR (100 MHz, DMSO-*d*_6_) δ 170.5 (CO); 152.9 (7-C); 150.6 (**C**_4_H_3_O); 140.0 (9-C); 136.3 (2-C); 131.9 (10-C); 125.4 (d, *J* = 32.9, 3-C); 123.6 (**C_4_H_3_**O); 122.7 (6-CH); 121.5 (5-CH); 119.5 (d, *J* = 273.8, CF_3_); 113.6 (**C_4_H_3_**O); 97.9 (8-CH); 43.7 (2 × CH_2_); 42.1 (2 × CH_2_). HRMS (ESI) calculated for C_18_H_16_F_3_N_4_O_4_^+^ [M+H]^+^ 409.1118, found 409.1078.

#### 3.1.20. 6-Fluoro-2-(furan-2-carbonyl)-7-(piperazin-1-yl)-3-trifluoromethylquinoxaline 1,4-Dioxide Hydrochloride (**16b**)

This compound was prepared from **9b** and *N*-Boc-piperazine as described for **12a**. R_f_ = 0.52 (chloroform-acetone, 5:1). A yellow powder, yield 47%, mp 246–248 °C (dec.). HPLC (LW = 300 nm, gradient B 20/80% (45 min)) t_R_ = 12.0 min, purity 95.5%. λ_max._, EtOH: 216, 276, 293, 338, 387, 418 nm. ^1^H-NMR (400 MHz, DMSO-*d_6_*) δ 9.58 (2H, br. s, NH_2_^+^); 8.33 (1H, d, *J* = 12.9, H-5); 8.23 (1H, br. m, C_4_H_3_O); 7.86 (1H, br. m, C_4_H_3_O); 7.73 (1H, d, *J* = 8.2, H-8); 6.83–6.82 (1H, m, C_4_H_3_O); 3.61 (4H, br. m, CH_2_); 3.29 (4H, br. m, CH_2_). ^13^C-NMR (100 MHz, DMSO-*d_6_*) δ 170.1 (CO) 157.7 (d, *J* = 260.7, 6-CF); 150.9 (**C_4_H_3_**O); 150.1 (**C**_4_H_3_O); 144.9 (d, *J* = 11.5, 7-C); 136.7 (2-C); 136.1 (d, *J* = 4.6, 9-C); 134.3 (d, *J* = 12.3, 10-C); 130.1 (d, *J* = 29.1, 3-C); 124.1 (**C_4_H_3_**O); 119.1 (d, *J* = 241.5, CF_3_); 113.6 (**C_4_H_3_**O); 107.1 (d, *J* = 30.5, 5-CH); 106.1 (d, *J* = 4.6, 8-CH); 46.4 (2 × CH_2_); 42.3 (2 × CH_2_). HRMS (ESI) calculated for C_18_H_15_F_4_N_4_O_4_^+^ [M+H]^+^ 427.1024, found 427.1030.

#### 3.1.21. 6-Chloro-2-(furan-2-carbonyl)-7-(piperazin-1-yl)-3-trifluoromethylquinoxaline 1,4-Dioxide Hydrochloride (**16c**)

This compound was prepared from **9c** and *N*-Boc-piperazine as described for **12a**. A yellow powder, yield 82%, mp > 260 °C (dec.). HPLC (LW = 300 nm, gradient B 10/90% (45 min)) t_R_ = 13.0 min, purity 96.4%. λ_max._, EtOH: 291, 374, 499 nm. ^1^H-NMR (400 MHz, DMSO-*d*_6_) δ 8.53 (1H, s, H-5); 8.14 (1H, br. m, C_4_H_3_O); 7.83 (1H, s, H-8); 7.76 (1H, br. m, C_4_H_3_O); 6.80–6.79 (1H, m, C_4_H_3_O); 3.45 (4H, br. m, CH_2_); 3.31 (4H, br. m, CH_2_). ^13^C-NMR (100 MHz, DMSO-*d*_6_) δ 170.4 (CO); 153.5 (7-C); 151.5 (**C_4_H_3_**O); 150.4 (**C**_4_H_3_O); 138.8 (9-C); 136.9 (10-C); 135.5 (d, *J* = 3.8, 2-C); 135.4 (6-CCl); 128.4 (q, *J* = 32.9, 3-C); 124.8 (**C_4_H_3_**O); 122.2 (5-CH); 121.1 (q, *J* = 265.1, CF_3_); 114.2 (**C_4_H_3_**O); 92.3 (8-CH); 47.7 (2 × CH_2_); 43.1 (2 × CH_2_). HRMS (ESI) calculated for C_18_H_15_ClF_3_N_4_O_4_^+^ [M+H]^+^ 443.0728, found 443.0734.

#### 3.1.22. 6,7-(Dipiperazin-1-yl)-2-(furan-2-carbonyl)-3-trifluoromethylquinoxaline 1,4-Dioxide Dihydrochloride (**16d**)

This compound was isolated as a minor component in the synthesis of the derivative **16b** described in 3.1.20. R_f_ = 0.36 (chloroform-acetone, 5:1). An orange powder, yield 25%, mp > 260 °C (dec.). HPLC (LW = 335 nm, gradient B 10/90% (45 min)) t_R_ = 8.55 min, purity 98.0%. λ_max._, EtOH: 209, 247, 274, 332, 442 nm. ^1^H-NMR (400 MHz, DMSO-*d*_6_) δ 8.66 (4H, br. s, NH_2_^+^); 8.20–8.17 (1H, br. m, C_4_H_3_O); 7.80–7.79 (1H, m, C_4_H_3_O); 7.63 (1H, s, H-8); 7.48 (1H, s, H-5); 6.81–6.79 (1H, br. m, C_4_H_3_O); 3.98–3.92 (4H, br. m, CH_2_); 3.53–3.51 (4H, br. m, CH_2_); 3.36–3.26 (8H, br. m, CH_2_). ^13^C-NMR (100 MHz, DMSO-*d*_6_) δ 170.8 (CO) 150.5 (**C_4_H_3_**O); 148.3 (7-C); 148.2 (9-C); 147.3 (6-C); 147.3 (2-C); 134.4 (10-C); 133.9 (**C_4_**H_3_O); 125.9 (d, *J* = 32.2, 2-C); 123.5 (**C_4_H_3_**O); 119.6 (d, *J* = 273.8, CF_3_); 113.6 (**C_4_H_3_**O); 103.4 (8-CH); 102.2 (5-CH); 53.3 (2 × CH_2_); 48.8 (4 × CH_2_); 48.6 (4 × CH_2_). HRMS (ESI) calculated for C_22_H_24_F_3_N_6_O_4_^+^ [M+H]^+^ 493.1806, found 493.1803.

#### 3.1.23. 7-(Piperazin-1-yl)-2-(thiophene-2-carbonyl)-3-trifluoromethylquinoxaline 1,4-Dioxide Hydrochloride (**17a**)

This compound was prepared from **10a** and *N*-Boc-piperazine as described for **12a**. An orange powder, yield 87%, mp 210–212 °C (dec.). HPLC (LW = 470 nm, gradient B 20/60/70% (43 min)) t_R_ = 16.72 min, purity 94.9%. λ_max._, EtOH: 235, 274, 301, 339, 383, 462 nm. ^1^H-NMR (400 MHz, DMSO-*d*_6_) δ 9.65 (2H, br. s, NH_2_^+^); 8.38 (1H, d, *J* = 9.4, H-5); 8.25 (1H, d, *J* = 5.1, C_4_H_3_S); 8.18 (1H, d, *J* = 4.3, C_4_H_3_S); 7.89 (1H, d, *J* = 9.4, H-6); 7.51 (1H, s, H-8); 7.28 (1H, t, *J* = 4.3, C_4_H_3_S); 3.81 (4H, br. s, CH_2_); 3.24 (4H, br. s, CH_2_). ^13^C-NMR (100 MHz, DMSO-*d*_6_) δ 176.51 (CO) 152.91 (7-C); 140.97 (9-C); 140.07 (2-C); 138.05 (2′-CH); 137.98 (1′-C); 137.73 (4′-CH); 136.88 (10-C); 132.00 (3-C); 129.38 (3′-CH); 122.58 (6-CH); 121.9 (q, *J* = 274.5, CF_3_); 121.47 (5-CH); 43.64 (2 × CH_2_); 42.06 (2 × CH_2_). HRMS (ESI) calculated for C_18_H_16_F_3_N_4_O_3_^+^ [M+H]^+^ 425.0890, found 425.0890.

#### 3.1.24. 6-Fluoro-7-(piperazin-1-yl)-2-(thiophene-2-carbonyl)-3-trifluoromethylquinoxaline 1,4-Dioxide Hydrochloride (**17b**)

This compound was prepared from **10b** and *N*-Boc-piperazine as described for **12a**. R_f_ = 0.54 (chloroform-acetone, 5:1). A yellow powder, yield 59%, mp 226–228 °C (dec.). HPLC (LW = 330 nm, gradient B 20/60/70% (43 min)) t_R_ = 12.32 min, purity 94.4%. λ_max._, EtOH: 241, 337, 444 nm. ^1^H-NMR (400 MHz, DMSO-*d*_6_) δ 9.71 (2H, br. s, NH_2_^+^); 8.32 (1H, d, *J* = 12.9, H-5); 8.28 (1H, d, *J* = 4.0, C_4_H_3_S); 8.18 (1H, d, *J* = 4.0, C_4_H_3_S); 7.73 (1H, d, *J* = 8.2, H-8); 7.29 (1H, t, *J* = 4.0, C_4_H_3_S); 3.62 (4H, br. m, CH_2_); 3.39 (4H, br. m, CH_2_). ^13^C-NMR (100 MHz, DMSO-*d*_6_) δ 176.6 (CO) 158.1 (d, *J* = 259.2, 6-CF); 145.3 (d, *J* = 10.7, 7-C); 142.3 (9-C); 138.8 (2-C); 138.7 (1′-C); 138.4 (2′-CH); 137.2 (4′-CH); 134.7 (d, *J* = 10.7, 10-C); 129.9 (3′-CH); 127.9 (q, *J* = 33.7, 3-C); 119.6 (q, *J* = 272.2, CF_3_); 107.4 (d, *J* = 28.4, 5-CH); 106.5 (8-CH); 46.7 (2 × CH_2_); 42.7 (2 × CH_2_). HRMS (ESI) calculated for C_18_H_15_F_4_N_4_O_3_S^+^ [M+H]^+^ 443.0796, found 443.0774.

#### 3.1.25. 6-Chloro-7-(piperazin-1-yl)-2-(thiophene-2-carbonyl)-3-trifluoromethylquinoxaline 1,4-Dioxide Hydrochloride (**17c**)

This compound was prepared from **10c** and *N*-Boc-piperazine as described for **12a**. A yellow powder, yield 73%, mp 235–236 °C (dec.). HPLC (LW = 395 nm, gradient B 20/60% (45 min)) t_R_ = 19.84 min, purity 97.7%. λ_max._, EtOH: 243, 287, 345, 394 nm. ^1^H-NMR (400 MHz, DMSO-*d*_6_) δ 9.59 (2H, br. s, NH_2_^+^); 8.59 (1H, s, H-8); 8.29 (1H, d, *J* = 7.4, C_4_H_3_S); 8.20 (1H, d, *J* = 4.0, C_4_H_3_S); 7.85 (1H, s, H-5); 7.30 (1H, t, *J* = 4.0, C_4_H_3_S); 4.33 (4H, br. m, CH_2_); 3.31 (4H, br. m, CH_2_). ^13^C-NMR (100 MHz, DMSO-*d*_6_) δ 176.0 (CO) 152.7 (7-C); 140.8 (9-C); 138.4 (2-C); 138.5 (2′-CH); 138.1 (4′-CH); 137.1 (1′-C); 134.9 (10-C); 134.1 (6-CCl); 129.5 (3′-CH); 128.1 (q, *J* = 32.9, 3-C); 121.9 (5-CH); 117.8 (q, *J* = 273.8, CF_3_); 47.5 (2 × CH_2_); 42.6 (2 × CH_2_). HRMS (ESI) calculated for C_18_H_15_ClF_3_N_4_O_3_S^+^ [M+H]^+^ 459.0500, found 459.0511.

#### 3.1.26. 6,7-(Dipiperazin-1-yl)-2-(thiophene-2-carbonyl)-3-trifluoromethylquinoxaline 1,4-Dioxide Hydrochloride (**17d**)

This compound was isolated as a minor component in the synthesis of the derivative **17b** described in 3.1.24. R_f_ = 0.32 (chloroform-acetone, 5:1). An orange powder, yield 26%, mp 251–252 °C (dec.). HPLC (LW = 390 nm, gradient B 20/60/70% (43 min)) t_R_ = 18.24 min, purity 99.7%. λ_max._, EtOH: 234, 275, 298, 337, 391 nm. ^1^H-NMR (400 MHz, DMSO-*d*_6_) δ 9.70 (4H, br. m, 2 × NH_2_^+^); 8.26 (1H, d, *J* = 4.3, C_4_H_3_S); 8.14 (1H, d, *J* = 4.3, C_4_H_3_S); 7.80 (1H, s, H-8); 7.67 (1H, s, H-5); 7.27 (1H, t, *J* = 4.3, C_4_H_3_S); 3.56 (8H, br. m, 4 ×CH_2_); 3.36 (8H, br. m, 4 ×CH_2_). ^13^C-NMR (100 MHz, DMSO-*d*_6_) δ 176.8 (CO) 150.1 (7-C); 149.4 (6-C); 141.5 (9-C); 138.6 (2-C); 138.3 (2′-CH); 138.1 (1′-C); 135.7 (4′-CH); 135.6 (10-C); 129.9 (3′-CH); 127.5 (q, *J* = 32.2, 3-C); 120.1 (q, *J* = 223.9, CF_3_); 107.4 (8-CH); 106.8 (5-CH); 46.2 (2 × CH_2_); 46.0 (2 × CH_2_); 42.9 (br. s, 4 × CH_2_). HRMS (ESI) calculated for C_22_H_24_F_3_N_6_O_3_S ^+^ [M+H]^+^ 509.1577, found 509.1575.

#### 3.1.27. 2-(1-Naphthoyl)-7-(piperazin-1-yl)-3-trifluoromethylquinoxaline 1,4-Dioxide Hydrochloride (**18a**)

This compound was prepared from **11a** and *N*-Boc-piperazine as described for **12a**. An orange powder, yield 91%, mp > 260 °C (dec.). HPLC (LW = 470 nm, gradient B 20/60/70% (43 min)) t_R_ = 23.51 min, purity 97.8%. λ_max._, EtOH: 256, 289, 430 nm. ^1^H-NMR (400 MHz, DMSO-*d*_6_) δ 9.70 (2H, br. s, NH_2_^+^); 8.81 (1H, s, H-8); 8.41 (1H, d, *J* = 9.0, H-6); 8.09 (2H, br. m, C_10_H_7_); 8.04–8.01 (2H, m, C_10_H_7_); 7.90 (1H, d, *J* = 9.0, H-5); 7.71 (1H, t, *J* = 7.0, C_10_H_7_); 7.61 (1H, t, *J* = 7.0, C_10_H_7_); 7.49 (1H, br. m, C_10_H_7_); 3.81 (4H, br. m, CH_2_); 3.39 (4H, br. m, CH_2_). ^13^C-NMR (100 MHz, DMSO-*d*_6_) δ 184.5 (CO) 152.9 (7-C); 140.1 (9-C); 137.5 (2-C); 135.9 (1′-C); 132.7 (10-C); 132.4 (9′-C); 131.9 (10′-C); 131.6 (4′-CH); 129.7 (2′-CH); 129.6 (8′-CH); 129.2 (5′-CH); 127.9 (3′-CH); 127.4 (7′-CH); 122.9 (6′-CH); 122.6 (6-CH); 125.5 (q, *J* = 33.7, 3-C); 121.5 (5-CH); 120.9(q, *J* = 273.8, CF_3_); 97.8 (8-CH); 43.6 (2 × CH_2_); 42.1 (2 × CH_2_). HRMS (ESI) calculated for C_24_H_20_F_3_N_4_O_3_^+^ [M+H]^+^ 469.1482, found 469.1492.

#### 3.1.28. 6-Fluoro-2-(1-naphthoyl)-7-(piperazin-1-yl)-3-trifluoromethylquinoxaline 1,4-Dioxide Hydrochloride (**18b**)

This compound was prepared from **11b** and *N*-Boc-piperazine as described for **12a**. A yellow powder, yield 48%, mp > 260 °C (dec.). R_f_ = 0.44 (chloroform-acetone, 5:1). HPLC (LW = 393 nm, gradient B 20/60/70% (43 min)) t_R_ = 26.09 min, purity 100.0%. λ_max._, EtOH: 256, 285, 298, 341, 393 nm. ^1^H-NMR (400 MHz, DMSO-*d*_6_) δ 9.69 (2H, br. s, NH_2_^+^); 8.83 (1H, s, H-8); 8.38 (1H, d, *J* = 12.5, H-5); 8.11 (2H, br. m, C_10_H_7_); 8.04 (2H, t, *J* = 8.6, C_10_H_7_); 7.74–7.72 (2H, m, C_10_H_7_); 7.64 (1H, t, *J* = 7.0, C_10_H_7_); 3.62 (4H, br. m, CH_2_); 3.29 (4H, br. m, CH_2_). ^13^C-NMR (100 MHz, DMSO-*d*_6_) δ 184.6 (CO) 158.1 (d, *J* = 158.1, 6-CF); 145.4 (7-C); 137.8 (9-C); 137.2 (2-C); 136.4 (1′-C); 134.9 (q, *J* = 20.7, 3-C); 134.6 (10-C); 133.2 (4′-CH); 132.8 (9′-C); 131.9 (10′-C); 130.2 (2′-CH); 130.1 (8′-CH); 129.7 (5′-CH); 128.4 (3′-CH); 127.9 (7′-CH); 123.4 (6′-CH); 119.8 (q, *J* = 273.8, CF_3_); 107.5 (d, *J* = 29.1, 5-CH); 106.4 (8-CH); 46.7 (2 × CH_2_); 42.7 (2 × CH_2_). HRMS (ESI) calculated for C_24_H_19_F_4_N_4_O_3_^+^ [M+H]^+^ 487.1388, found 487.1414.

#### 3.1.29. 6-Chloro-2-(1-naphthoyl)-7-(piperazin-1-yl)-3-trifluoromethylquinoxaline 1,4-Dioxide Hydrochloride (**18c**)

This compound was prepared from **11c** and *N*-Boc-piperazine as described for **12a**. A yellow powder, yield 64%, mp > 260 °C (dec.). HPLC (LW = 395 nm, gradient B 20/60/70% (43 min)) t_R_ = 22.29 min, purity 94.6%. λ_max._, EtOH: 256, 287, 298, 348, 395 nm. ^1^H-NMR (400 MHz, DMSO-*d*_6_) δ 9.70 (2H, br. s, NH_2_^+^); 8.82 (1H, s, H-8); 8.61 (1H, s, H-5); 8.10 (2H, br. m, C_10_H_7_); 8.05–7.99 (3H, m, C_10_H_7_); 7.72 (1H, t, *J* = 7.4, C_10_H_7_); 7.62 (1H, t, *J* = 7.4, C_10_H_7_); 3.49 (4H, br. m, CH_2_); 3.28 (4H, br. m, CH_2_). ^13^C-NMR (100 MHz, DMSO-*d*_6_) δ 184.5 (CO) 153.2 (7-C); 149.5 (9-C); 147.4 (2-C); 138.4 (1′-C); 135.9 (10-C); 134.9 (9′-C); 134.2 (10′-C); 132.8 (6-CCl); 132.3 (4′-CH); 131.9 (q, *J* = 37.6, 3-C); 131.4 (5′-CH); 129.8 (2′-CH); 129.6 (8′-CH); 129.3 (3′-CH); 127.9 (7′-CH); 127.5 (6′-CH); 122.9 (5-CH); 120.7 (q, *J* = 263.0, CF_3_); 108.9 (8-CH); 47.9 (2 × CH_2_); 43.7 (2 × CH_2_). HRMS (ESI) calculated for C_24_H_19_ClF_3_N_4_O_3_^+^ [M+H]^+^ 503.1092, found 503.1071.

#### 3.1.30. 2-(1-Naphthoyl)-7-(dipiperazin-1-yl)-3-trifluoromethylquinoxaline 1,4-Dioxide Hydrochloride (**18d**)

This compound was isolated as a minor component in the synthesis of the derivative **18b** described in 3.1.28. R_f_ = 0.25 (chloroform-acetone, 5:1). An orange powder, yield 32%, mp > 260 °C (dec.). HPLC (LW = 440 nm, gradient B 20/60/70% (43 min)) t_R_ = 19.46 min, purity 98.8%. λ_max._, EtOH: 256, 299, 342, 440 nm. ^1^H-NMR (400 MHz, DMSO-*d*_6_) δ 9.64 (2H, br. s, NH_2_^+^); 9.59 (2H, br. s, NH_2_^+^); 8.78 (1H, s, H-8); 8.08 (2H, br. m, C_10_H_7_); 8.02 (2H, t, *J* = 7.8, C_10_H_7_); 7.84 (1H, s, H-5); 7.71 (1H, t, *J* = 7.8, C_10_H_7_); 7.65 (1H, br. m, C_10_H_7_); 7.61 (1H, t, *J* = 7.8, C_10_H_7_); 3.54 (8H, br. m, 4 × CH_2_); 3.29 (8H, br. m, 4 × CH_2_). ^13^C-NMR (100 MHz, DMSO-*d*_6_) δ 184.9 (CO) 150.1 (7-C); 149.4 (6-C); 136.6 (9-C); 136.3 (2-C); 135.7 (1′-C); 135.6 (10-C); 133.2 (4′-CH); 132.8 (9′-C); 132.1 (10′-C); 130.2 (2′-CH); 130.1 (8′-CH); 129.6 (5′-CH); 128.4 (3′-CH); 127.9 (q, *J* = 32.2, 3-C); 127.8 (7′-CH); 123.4 (6′-CH); 119.9 (q, *J* = 272.9, CF_3_); 107.5 (5-CH); 106.7 (8-CH); 46.2 (2 × CH_2_); 46.0 (2 × CH_2_); 42.9 (br. m, 4 × CH_2_). HRMS (ESI) calculated for C_28_H_28_F_3_N_6_O_3_^+^ [M+H]^+^ 553.2169, found 553.2147.

### 3.2. Biology

#### 3.2.1. Microbial Cultures and Growth Conditions

All the bacterial cultures used in this study are summarized in Table 3.

The rest of the test strains: *Staphylococcus aureus* ATCC 29213, *Staphylococcus epidermidis* ATCC 14990, *Enterococcus faecalis* ATCC 29212, *Escherichia coli* ATCC 25922, *Klebsiella pneumoniae* 1951, *Pseudomonas aeruginosa* ATCC 27853, *Candida albicans* ATCC 10231, *Microsporum canis* B-200 were obtained from the culture collection of the Antibiotic Science Center (Moscow, Russian Federation). Strains were stored at −75 °C in trypticase soy broth with 15% glycerol. Before the experiment, the bacterial strains were plated on trypticase soy agar, the fungus strains were plated on Sabouraud dextrose agar and incubated at 36 °C.

*Mycobacterium* strains were grown in liquid Middlebrook 7H9 medium (Himedia, Mumbai, India) supplemented with oleic albumin dextrose catalase (OADC, Himedia), 0.1% Tween-80 (*v*/*v*) and 0.4% glycerol (*v*/*v*), while the M290 Soyabean Casein Digest Agar (Himedia) and Middlebrook 7H11 agar supplemented with 0.2% glycerol (*v*/*v*) and OADC were used as solid media for *M. smegmatis* and *M. tuberculosis* growth, respectively. *Escherichia coli* DH5α was used for plasmids propagation and was grown in liquid or on agarized LB medium with the addition of the corresponding selection antibiotics, when required. Cultures in liquid medium were incubated in the Multitron incubator shaker (Infors HT, Bottmingen-Basel, Switzerland) at 37 °C and 250 rpm.

The remaining bacterial and fungi strains were stored at −75 °C in trypticase soy broth with 15% glycerol. Before the experiment, the bacterial strains were plated on trypticase soy agar, while the fungi strains were plated on Sabouraud dextrose agar and incubated at 36 °C.

#### 3.2.2. Minimal Inhibitory Concentrations Determination

##### MIC Determination on *M. smegmatis* Strains

Minimal inhibitory concentrations (MICs) of the studied compounds on *M. smegmatis* were determined in liquid medium by the serial two-fold dilution method, as described before [53]. Briefly, *M. smegmatis* strains were cultured overnight in 7H9 medium, then diluted in the proportion of 1:200 in fresh medium (to approximately OD_600_ = 0.05), and 196 μL of the diluted culture were poured in sterile non-treated 96-well flat-bottom culture plates (Eppendorf, Hamburg, Germany) and 4 μL of serial two-fold dilutions of the tested compounds in DMSO were added to the wells to final concentrations ranging from 1 to 32 μg/mL (a total of six concentrations), while 4 μL of DMSO were used as untreated control. The plates were incubated at 37 °C and 250 rpm for 48 h. The experiments were carried out in triplicates, the MIC was determined as the lowest concentration of the compound with no visible bacterial growth.

##### MIC Determination on *M. tuberculosis* Strains

Serial dilutions of drug containing solutions and autoluminescent *M. tuberculosis* AlRa or UAlRv strains broth culture (OD_600_ of 0.3 to 1.0) were prepared. RLU counts from the same batch of triplicate samples were measured according to the designed time points. Ninety-six-well plates were measured as rendering the amount of light by using a Veritas™ Microplate Luminometer Operating Manual (Promega, Madison, WI, USA). MIC_lux_ was determined as the lowest concentration that can inhibit > 90% RLUs compared with that from the untreated controls. The MIC_lux_ of autoluminescent strains were determined by detecting RLUs from these samples as described before [51,52]. The concentrations used were twofold series of dilutions for this purpose, ranging from 0.078 to 20 μg/mL (a total of nine concentrations). The light from each 1.5 mL tube containing total 500 μL culture was detected using GloMax 20/20^n^ (Promega, Madison, WI, USA). Each above experiment performed in triplicate (three independent experiments) or more by independent persons and from three or more independent cultures.

##### MIC Determination on the Rest of the Test Strains

The determination of antimicrobial activity in vitro was performed by microdilution in Mueller Hinton broth (BD BBL™, 38800 Le Pont-de-Claix, France) for bacterial strains and in RPMI 1640 medium (Sigma-Aldrich), supplemented with 2% glucose for fungi, in accordance with the recommendations of the CLSI [54,55,56]. The optical density of the inoculum of each strain was measured on a DIN-1 densitometer (Biosan, Rīga, Latvia) and adjusted the microorganisms’ density of 0.50 McFarland. Bacterial cells were diluted in Mueller Hinton broth toa final concentration of 10^5^ CFU/mL, while *Candida* was diluted in RPMI 1640 medium to a final concentration of 10^3^ CFU/mL. The filamentous fungus *M. canis* was incubated until the formation of aerial mycelium (about 5–10 days). Conidia and spores were collected by swab and transferred to physiological saline. The inoculum concentration was counted by hemocytometer and diluted in RPMI 1640 medium to a final concentration of 10^3^ CFU/mL. The stock solutions (10 mg/mL) of the test compound were prepared in DMSO, and were used to prepare the serial twofold dilutions with final concentrations ranging from 0.25 to 32 μg/mL (a total of 8 concentrations). The experimental plates were incubated at 36 °C, with the time of incubation depending on the microorganism: 18–20 h for bacterial strains, 24–48 h for *Candida*, and 72–96 h for *M. canis*. The experiments were carried out in triplicates, the MIC was defined as the lowest concentration of a compound at which there was no visible growth of microorganisms.

#### 3.2.3. Obtaining Spontaneous Drug-Resistant *M. smegmatis* Mutants

Spontaneous *M. smegmatis* mutants resistant to the tested aminoderivatives of 3-trifluoromethylquinoxaline 1,4-dioxide were obtained as described before [57,58]. Briefly, overnight *M. smegmatis* mc^2^ 155 cultures in the quantity of 1.4 × 10^8^ CFU were plated on M290 plates, supplemented with 4 × MICs of the compounds **12–18**. A total of 3 plates with each compound were used in the experiments. The plates were incubated at 37 °C until single colonies were visible (5–7 days). The colonies were counted to calculate the frequency of drug-resistance emergence. The colonies were stroked on drug-free M290 plates, and after that re-stroked on the plates with 4 × MIC of the respective compound to confirm the drug-resistance. Three mutants resistant to each of the compound were randomly selected for further analysis and whole-genomic sequencing.

#### 3.2.4. *M. smegmatis* Whole-Genomic Sequencing and Analysis

Mycobacterial genomic DNA was isolated from 10 mL by enzymatic lysis as described by Belisle et al. [59], after preliminary isolation, DNA was treated with RNase A (Thermo Fischer Scientific, Waltham, MA, USA) and extracted in the phenol-chloroform-isoamyl alcohol solution (25:24:1).

A total of 250 ng genomic DNA was taken for shotgun sequencing library preparation. After DNA sonication on Covaris S220 System (Covaris, Woburn, MA, USA), the size (400–500 b.p.) and quality of fragmented samples were assessed on Agilent 2100 Bioanalyzer (Agilent, Santa Clara, CA, USA) according to the manufacturer’s manual. NEBNext Ultra II DNA Library Prep Kit (New England Biolabs, Ipswich, MA, USA) was used for pair-ended library preparation, and NEBNext Multiplex Oligos kit for Illumina (96 Index Primers, New England Biolabs) was used for libraries’ indexing. The libraries were quantified by Q Ipswich, MA,uant-iT DNA Assay Kit, High Sensitivity (Thermo Fischer Scientific). DNA sequencing (2 × 125 b.p.) was performed on the HiSeq 2500 platform (Illumina, San Diego, CA, USA) according to the manufacturer’s recommendations.

The obtained reads’ quality was assessed with FastQC (v. 0.11.8) [60], as the quality was good, we proceeded to the assembly without trimming the reads. The reads were aligned to the reference genome (NC_008596.1, PRJNA57701) using the BWA-MEM algorithm [61]. The pileup was generated by mpileup (-B-f) in SAMtools [62]. Single nucleotide variants were called by running mpileup2snp (--min-avg-qual 30 --min-var-freq 0.80 --p-value 0.01 --output-vcf 1 --variants 1) in VarScan (v. 2.3.9) [63]. Annotation was created using vcf_annotate.pl (developed by Natalya Mikheecheva of the Laboratory of Bacterial Genetics, Vavilov Institute of General Genetics, Moscow, Russia). The non-synonymous single nucleotide variants found within open reading frames and absent in the wild-type strain were selected for further analysis. The raw sequencing data (SRA) are publicly available in NCBI GenBank (BioProject ID: PRJNA778794).

#### 3.2.5. Introduction of the Targeted Mutation in MSMEG_4883 Gene

The targeted mutation in *MSMEG_4883* gene of *M. smegmatis* mc^2^ 155 strain was introduced by homologous recombination, using the p2NIL/pGOAL19 suicide vector system [64]. The gene *MSMEG_4883* with flanking arms of approximately 1900 b.p., adjacent to the mutation site was amplified by the Q5 polymerase (New England Biolabs) with primers pN_4883_f (5′-TTTTAAGCTTACCGAGATCGCGTTGAACTT-3′) and pN_4883_r (5′-TTTTTTAATTAACCGACAAGGGAGTACACGAG-3′) with incorporated *Hind*III and *Pac*I restriction ezymes’ recognition sites, respectively. The amplified products were digested with respective restriction enzymes (Thermo Scientific) and ligated in the p2NIL plasmid. The cassette from pGOAL19 was subsequently cloned in the obtained plasmids at the *Pac*I restriction site. The plasmids were electroporated in *M. smegmatis* mc^2^ 155 cells as described earlier [65] and plated on M290 plates supplemented with kanamycin (50 μg/mL), hygromycin (50 μg/mL) and X-Gal (50 μg/mL), blue single-crossover colonies were selected. Blue colonies were grown overnight in liquid 7H9 medium with OADC, and serial 10-fold dilutions were plated on M290 plates supplemented with X-Gal (50 μg/mL) and sucrose (2% *w*/*v*), white double-crossover colonies were selected and tested for Km susceptibility. Target genes were then Sanger-sequenced for a final confirmation of the mutation using the primers pM_4883_f (5′-TTTTCTGCAGAGGAGGAATTGTTATGAGCATATCGCTGCTGC-3′) and s_4883_r (5′-GACCTTGGATCCGCCGTAG-3′). Primer-BLAST [66] was used for primer selection.

#### 3.2.6. Mechanism of Action Determination

Reporter strain *E. coli BW25113* -pDualrep2 was used as previously described [37]. A total of 1 µL of 20 mg/mL solution of each sample was applied to agar plate containing a lawn of the reporter strain. After overnight incubation at 37 °C, the plate was scanned by ChemiDoc (Bio-Rad Laboratories, 1000 Alfred Nobel Drive, Hercules, CA, USA) system with two channels including “Cy3-blot” (553/574 nm, green pseudocolor) for RFP fluorescence and “Cy5-blot” (588/633 nm, red pseudocolor) for Katushka2S fluorescence. Induction of expression of Katushka2S is triggered by translation inhibitors, while RFP is up regulated by induction of DNA damage SOS response. 1 µL of levofloxacin (50 μg/mL) and erythromycin (50 μg/mL) were used as positive controls for DNA biosynthesis and ribosome inhibitors, respectively.

## 4. Conclusions

We describe the synthesis and the profile of antimicrobial properties for the new series of 2-acyl-7-amino-3-trifluoromethylquinoxaline 1,4-dioxide derivatives. The series of 7-amino-3-trifluoromethylquinoxaline 1,4-dioxides was prepared by the nucleophilic substitution of halogen atoms which as in case of 6,7-dihalogeno-3-trifluoromethylquinoxaline 1,4-dioxides proceeds with a good regioselectivity in the position 7. Additionally, these compounds could be obtained by the Beirut reaction between 5-amino-6-halogensubstituted benzofuroxan and 1-acyl-3,3,3-trifluoroacetones which also has a high regioselectivity and leads to 2-acyl-7-amino-3-trifluoromethylquinoxaline 1,4-dioxides.

Biological screening of the synthesized 3-trifluoromethylquinoxaline 1,4-dioxides on a panel of bacterial strains and fungi, allowed us to identify several derivatives (**12c**, **13c**, **14c**, **15b**, **15c**, **16c**, **17c** and **18b**) with significant antimicrobial activity comparable to the reference drug, such as DIOX, RIF, AMPH and CIP. The structure-activity relationship revealed a number of important features for the ability of compounds to inhibit the growth of pathogen: the introduction of a halogen atom in the position 6 of the quinoxaline ring increased the antimicrobial activity of the compounds especially on Gram-positive species whereas acyl groups on the position 2 was not so decisive for biological activity. Though a relatively low activity was observed on *M. tuberculosis* strains, all the compounds except **16a**, **16d**, **17b**, **17d** and **18b–c** had MIC values (2–8 μg/mL) comparable to those of rifampicin (MIC = 4 μg/mL) on *M. smegmatis*
*mc*^2^ 155, suggesting these compounds as a basis for further development of antimycobacterial agents. Notably most of the obtained compounds were inactive against tested fungal pathogens such as *Candida albicans ATCC 10231* and *M. canis B*-200, with the exception of **15b** (MIC = 4–8 μg/mL compared to AMPH MIC = 0.75 and 2 μg/mL respectively). According to the received data the compound **13c** from this series stood out with remarkable activity against most of the tested strains with MIC values in the range of 0.25–10 μg/mL and could be perspective for further evaluation of chemotherapeutic properties and toxicity. It is worth noting, that some of the described compounds were able to inhibit proliferation of breast cancer cell line MCF-7 at submicromolar concentrations [67]. Thus, additional evaluation of cytotoxicity on non-cancer cell lines might be required for a further structure optimization of these derivatives during their development as selective anti-bacterial agents.

The whole-genomic sequences obtained for *M. smegmatis* spontaneous drug-resistant mutants revealed the high mutagenic potential of 7-aminoderivatives of 2-acyl-3-trifluoromethylquinoxaline 1,4-dioxide, suggesting the same mechanism of DNA-cleavage activity, as described for the previous quinoxaline 1,4-dioxides [42,43], but may still be a clue to a deeper understanding of their redox activation in mycobacterial cells and the mechanisms of drug-resistance emergence. Suggested mechanism of 3-trifluoromethylquinoxaline 1,4-dioxides is also confirmed by upregulation of SOS-response induced reporter pDualrep2: tested compounds and DIOX have demonstrated the same results as other well-studied DNA-damaging agents. However, the noticeable inhibition of protein synthesis was observed for derivatives **13a–c**, as well as the absence of correlations between antibacterial activity and the SOS response of bacterial cells for several compounds (**12b**, **15d**, **18d)**, showing that a more detailed evaluation of the mechanism of action of these compounds and the identification of their intracellular targets are necessary.

The described compounds can be suggested for further development as potential drug candidates with selective antibacterial activity for Gram-positive bacteria, and especially *Mycobacterium* species, based on structural optimization of this promising scaffold.

## Data Availability

Data is contained within the article or Appendix A.

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
