# Peer review of "Synthesis and Characterization of Novel 2-Acyl-3-trifluoromethylquinoxaline 1,4-Dioxides as Potential Antimicrobial Agents"

_pharmaceuticals, 2022, doi:10.3390/ph15020155_

Round 1

Reviewer 1 Report

Authors report “Synthesis and characterization of novel 2-acyl-3-trifluoromethylquinoxaline 1,4-dioxides as po-tential antimicrobial agents” The work will be useful in drug design and development. However, it needs minor revision as suggested below before publication. Comments 1 In gram-negative or gram-positive, gram is the name of scientist it should be written as Gram. 2 In the abstract, precise results should be described. 3 Except few, most of the references cited in the introduction are old, authors should quote latest references. 4 There is no introduction or research background on yeast (Candida albicans), although authors tested their compounds against Candida albicans. Relevant literature should be added with significance of testing. 5 2D NMR techniques are needed to better characterize the synthesized structures (HSQC at least) so authors should provide 2D NMR spectra of few compounds, especially the compounds which are more potent. 6 How many concentrations were tested to calculate the MIC and how many replicate were used? 7 Abstracts should be revised and improved.

Author Response

We would like to thank the reviewers for carefully reading our manuscript and providing valuable criticism. Following the reviewers comments, we have accomplished additional experiments and added the required corrections in the text and supplemental materials.

  1. In gram-negative or gram-positive, gram is the name of scientist it should be written as Gram.

Response: Thank you for your suggestions. The remark was taken into account and corrected throughout the manuscript.

  1. In the abstract, precise results should be described.

Response: The key SAR results and the discussion of drug-resistance mechanism have been also added to the abstract.

  1. Except few, most of the references cited in the introduction are old, authors should quote latest references.

Response: The references in the text were thoroughly revised and updated.

  1. There is no introduction or research background on yeast (Candida albicans), although authors tested their compounds against Candida albicans. Relevant literature should be added with significance of testing.

Response: Thank you for your suggestion. The background for antifungal studies of quinoxalin 1,4-dioxides and relevant reference were added into the Introduction.

  1. 2D NMR techniques are needed to better characterize the synthesized structures (HSQC at least) so authors should provide 2D NMR spectra of few compounds, especially the compounds which are more potent.

Response: Thank you for your suggestion. 2D NMR spectra (HSQC, HMBC, CIGAR-HMBC) for the selected compounds 13b and 13c which confirmed assigned structures were added in the text and the Supplementary Materials.

  1. How many concentrations were tested to calculate the MIC and how many replicate were used?

Response: The number of concentrations and replicates were added into the Experimental Section of the manuscript.

  1. Abstracts should be revised and improved.

Response: The abstract was revised and improved as stated above.

Reviewer 2 Report

This manuscript describes the synthesis and antimicrobial evaluation of quinoxaline dioxide derivatives. The overall concept is good, but I have two major issues that have to be resolved before the evaluation process can continue:

Major issues:

  1. Firstly, I am not very convinced about the claimed regioselectivity of the substitution of 6,7-dihalogen intermediates. Authors claim that there is a regioselectivity for substitution of halogen in position 7. The proof is based on comparison of NMR shifts. I feel this is not enough, especially when the claimed regioselectivity is in contradiction with data in reference [31], as the authors themselves stated correctly. I would require at least some 2D NMR experiments (for selected representative compounds), which should be available with the NMR instrumentation available to the authors. Optionally, X-ray crystallography should give the answer. According to the authors, only the 6,7-diF (and not 6,7-diCl) intermediates underwent the nucleophilic substitution of both halogens at the same time. This requires explanation because chlorine should be a better leaving group in comparison to fluorine, so why this double substitution was not observed for 6,7-diCl intermediate. Incorrect names of compounds 3.1.28 and 3.1.29 also add up to the confusion. It looks slike the authors themselves might not be sure in the beginning to which position the nucleophile goes.
  1. The second major issue is the missing data on cytotoxicity of the final compounds. Or, better said, the commented data. At least some of the compounds of this manuscript are covered in a patent of the same authors (RU2746395 C1), and these compounds were filed as ‘compounds inhibiting tumour cells’, with IC 50 values in the low micromolar range. Therefore it is completely honest to declare these compounds as antimicrobial candidates when it is clear that there will be serious issues with the selectivity. The commentary about cytotoxicity data must be included in the manuscript.

Suggestions:

  • From medchem point of view, the final compounds are structurally similar to antibacterial fluoroquinolones (chelating groups at position 1 and 2), fluorine and piperazinyl substituents at positions 6 and 7 – the similarity would d be even higher if the regioselectivity of the nucleophilic substitution is inversed to what has been claimed). The authors can comment on whether or not they think that some part of the MoA could be due to the fluoroquinolone similarity.

Minor issues:

  • in schemes, the designation of substituent R1, R2 etc. The number should be upper script.
  • Scheme 1 – title. 3-aсyl-1,1,1-trifluoroacetone is not a correct name.
  • for other minor issues, please see the enclosed PDF directly

Author Response

We would like to thank the reviewers for carefully reading our manuscript and providing valuable criticism. Following the reviewers comments, we have accomplished additional experiments and added the required corrections in the text and supplemental materials.

  1. Firstly, I am not very convinced about the claimed regioselectivity of the substitution of 6,7-dihalogen intermediates. Authors claim that there is a regioselectivity for substitution of halogen in position 7. The proof is based on comparison of NMR shifts. I feel this is not enough, especially when the claimed regioselectivity is in contradiction with data in reference [31], as the authors themselves stated correctly. I would require at least some 2D NMR experiments (for selected representative compounds), which should be available with the NMR instrumentation available to the authors. Optionally, X-ray crystallography should give the answer.

Response: Thank you for your suggestion. 2D NMR spectra (HSQC, HMBC, CIGAR-HMBC) for the selected compounds 13b and 13c which confirmed assigned structures were added in the text and the Supplementary Materials.

  1. According to the authors, only the 6,7-diF (and not 6,7-diCl) intermediates underwent the nucleophilic substitution of both halogens at the same time. This requires explanation because chlorine should be a better leaving group in comparison to fluorine, so why this double substitution was not observed for 6,7-diCl intermediate.

Response: The nature of halogen atom affects the rate of nucleophilic substitution. An increase in the electronegativity of halogen causes a decrease in the electron density at the site of attack, resulting in a faster substitution of halogen with a nucleophile. Thus, in the aromatic nucleophilic substitution, when fluorine is the halogen atom, the relative rate is ~3300 (compared with I = 1). The fact that fluorine is the most reactive among the halogens in aromatic nucleophilic substitutions SN2Ar is a good evidence that this mechanism is different from the SN1 and the SN2, where fluoro is the poorest leaving group among of the halogens [Senger, N. A., Bo, B., Cheng, Q., Keeffe, J. R., Gronert, S., & Wu, W. (2012). The Element Effect Revisited: Factors Determining Leaving Group Ability in Activated Nucleophilic Aromatic Substitution Reactions. The Journal of Organic Chemistry, 77(21), 9535–9540. doi:10.1021/jo301134q].

  1. Incorrect names of compounds 3.1.28 and 3.1.29 also add up to the confusion. It looks like the authors themselves might not be sure in the beginning to which position the nucleophile goes.

Response: We apologize for this mistake: the names of compounds have been checked and corrected.

  1. The second major issue is the missing data on cytotoxicity of the final compounds. Or, better said, the commented data. At least some of the compounds of this manuscript are covered in a patent of the same authors (RU2746395 C1), and these compounds were filed as ‘compounds inhibiting tumour cells’, with IC50 values in the low micromolar range. Therefore it is completely honest to declare these compounds as antimicrobial candidates when it is clear that there will be serious issues with the selectivity. The commentary about cytotoxicity data must be included in the manuscript.

Response: The commentary about the cytotoxicity of the described compounds and the corresponding reference were added to the Conclusion section.

  1. From medchem point of view, the final compounds are structurally similar to antibacterial fluoroquinolones (chelating groups at position 1 and 2), fluorine and piperazinyl substituents at positions 6 and 7 – the similarity would d be even higher if the regioselectivity of the nucleophilic substitution is inversed to what has been claimed). The authors can comment on whether or not they think that some part of the MoA could be due to the fluoroquinolone similarity.

Response: Thank you for your suggestion. Despite the obvious similarity of fluoroquinolones with the obtained derivatives of quinoxaline 1,4-dioxide, the evaluation of some early synthesized piperazine-substituted quinoxaline 1,4-dioxides [19, 29] showed that such compounds do not bind to DNA and not have topoisomerases inhibition activity (not published data), which may indicate a different from fluoroquinolones mode of action, namely, characteristic for quinoxaline 1,4-dioxides ROS-mediated DNA damage.

  1. In schemes, the designation of substituent R1, R2 etc. The number should be upper script.

Response: Thank you for your suggestion. The designation of substituents in Schemes and Figures were corrected.

  1. Scheme 1 – title. 3-aсyl-1,1,1-trifluoroacetone is not a correct name.

Response: The name of started 1,3-dicarbonyl compound in the Scheme 1 has been changed.

  1. The sentence starts with 'currently' but cites a publication from 1978. I believe that such old publications should not be used for such an up-to-date topic as antimicrobial research.

Response: All references relating to the old publications were updated.

  1. This text sounds like you are substituting fluorine in CF3. But it is not the case. Pls rewrite so it is clear you are substituting the halogen on the quinoxaline core: «However, attempts to carry out the nucleophilic substitution of the halogen atoms in 2-acyl-3-trifluoromethylquinoxaline 1,4-dioxides 5-11а-с by piperazine in DMF as described earlier [17] were ineffective due to the formation of a mixture of deoxygenated analogs as the main products of the reaction». Same as above: Nevertheless, the use of N-Boc-piperazine in this reaction was found to be more effective and reduced side products of the halogens substitution in 3-trifluoromethylquinoxaline 1,4-dioxides 5-11.

Response: These sentences was thoroughly revised and corrected as suggested.

Reviewer 3 Report

The manuscript titled “ Synthesis and characterization of novel 2-acyl-3-trifluoromethylquinoxaline 1,4-dioxides as potential antimicrobial agents” written by Buravchenko et al. describes the synthesis of new quinoxaline 1,4-dioxide derivatives.and their biological evaluation against panel of bacterial strains, including gram-positive and gram-negative bacteria, and fungi such as C. albicans ATCC 10231 and M. canis B-200. The manuscript is clearly written and well organized and, then, it can be accepted for the publication in the journal after few corrections.

Details:

  • Introduction: The bibliography needs an update, replace the old reference (1978!) with the recent ChemMedChem, 2021, 16(1), 65–80.

The sentence “Gram-negative bacteria, such as Klebsiella pneumoniae and Escherichia coli, can cause severe and even fatal infections, being a particular threat to people with weak or immature immunity, such as the newborns, the elderly, and patients with oncology or AIDS.” needs a reference (e.g. Front. Public Health, https://doi.org/10.3389/fpubh.2019.00151 2019, 7, 151.

Analogously, after the sentence “A particularly alarming tendency … with any of the existing antimicrobial drugs” add the recent reference Future Medicinal Chemistry, 2021, 13(6), 529–531

  • The trifluoromethyl group is one of the privileged structural fragments in medicinal chemistry which used for modulating of the pharmacological properties of molecules due to its unique characteristic such as molecular volume, lipophilicity, and the ability to form hydrogen bonds [15,16]”: this sentence is too general, it should be better to report some examples of antimicrobial agents in which the introduction of a trifluoromethyl group improved the activity.
  • Page 5, Scheme 2: Correct “Synthesis of 7-(piperazin-1-yl)-3-trifluoromethylquinoxaline 1,4-dioxides 12-14a-c and 15-19a-c” in “Synthesis of 7-(piperazin-1-yl)-3-trifluoromethylquinoxaline 1,4-dioxides 12-19a-c”.
  • Page 9,. Mechanism of action determination : Authors should clarify the reasons that led to investigate the mechanism of action of the new compounds “by means of the pDualrep2 reported system”.

Author Response

We would like to thank the reviewers for carefully reading our manuscript and providing valuable criticism. Following the reviewers comments, we have accomplished additional experiments and added the required corrections in the text and supplemental materials.

  1. Introduction: The bibliography needs an update, replace the old reference (1978!) with the recent ChemMedChem, 2021, 16(1), 65–80.

The sentence “Gram-negative bacteria, such as Klebsiella pneumoniae and Escherichia coli, can cause severe and even fatal infections, being a particular threat to people with weak or immature immunity, such as the newborns, the elderly, and patients with oncology or AIDS.” needs a reference (e.g. Front. Public Health, https://doi.org/10.3389/fpubh.2019.00151 2019, 7, 151.

Analogously, after the sentence “A particularly alarming tendency … with any of the existing antimicrobial drugs” add the recent reference Future Medicinal Chemistry, 2021, 13(6), 529–531.

Response: Thank you for your suggestion. The marked reference has been changed as recommended.

  1. “The trifluoromethyl group is one of the privileged structural fragments in medicinal chemistry which used for modulating of the pharmacological properties of molecules due to its unique characteristic such as molecular volume, lipophilicity, and the ability to form hydrogen bonds [15,16]”: this sentence is too general, it should be better to report some examples of antimicrobial agents in which the introduction of a trifluoromethyl group improved the activity.

Response: We have now examples with relevant citations, as suggested.

  1. Page 5, Scheme 2: Correct “Synthesis of 7-(piperazin-1-yl)-3-trifluoromethylquinoxaline 1,4-dioxides 12-14a-c and 15-19a-c” in “Synthesis of 7-(piperazin-1-yl)-3-trifluoromethylquinoxaline 1,4-dioxides 12-19a-c”.

Response: Corrected as suggested.

  1. Page 9, Mechanism of action determination : Authors should clarify the reasons that led to investigate the mechanism of action of the new compounds “by means of the pDualrep2 reported system”.

Response: We have now added a sentence, explaining why the pDualrep2 system was used in this study, as suggested: “As it has been previously shown, that DIOX induces SOS-response in bacterial cells [Mazanko, M.S.; Chistyakov, V.A.; Prazdnova, E.V.; Pokudina, I.O.; Churilov, M.N.; Chmyhalo, V.K.; Batyushin, M.M. Dioxidine induces bacterial resistance to antibiotics. Mol. Gen. Microbiol. Virol. 2016, 31, 227-232], the mechanism…”

The English style and grammar have been also edited by a native speaker throughout the manuscript. We are now submitting the revised manuscript with changes highlighted in green. I hope that the revised manuscript is now acceptable for publication.

Thank you again all reviewers for valuable suggestions and a consideration of our work.

Sincerely,

Andrey E. Shchekotikhin and co-authors

Round 2

Reviewer 2 Report

The authors have sufficiently adres sed all major issues raised in my previous report.